# Causal Reasoning and Large Language Models: Opening a New Frontier for Causality

**Emre Kıcıman** *
*Microsoft Research*
emrek@microsoft.com

**Robert Osazuwa Ness**
*Microsoft Research*
robertness@microsoft.com

**Amit Sharma**
*Microsoft Research*
amshar@microsoft.com

**Chenhao Tan**
*University of Chicago*
chenhao@uchicago.edu

**Reviewed on OpenReview:** *https://openreview.net/forum?id=mqoxLkX210*

## Abstract

The causal capabilities of large language models (LLMs) are a matter of significant debate, with critical implications for the use of LLMs in societally impactful domains such as medicine, science, law, and policy. We conduct a "behavorial" study of LLMs to benchmark their capability in generating causal arguments. Across a wide range of tasks, we find that LLMs can generate text corresponding to correct causal arguments with high probability, surpassing the best-performing existing methods. Algorithms based on GPT-3.5 and 4 outperform existing algorithms on a pairwise causal discovery task (97%, 13 points gain), counterfactual reasoning task (92%, 20 points gain) and event causality (86% accuracy in determining necessary and sufficient causes in vignettes). We perform robustness checks across tasks and show that the capabilities cannot be explained by dataset memorization alone, especially since LLMs generalize to novel datasets that were created after the training cutoff date.

That said, LLMs exhibit unpredictable failure modes and we discuss the kinds of errors that may be improved and what are the fundamental limits of LLM-based answers. Overall, by operating on the text metadata, LLMs bring capabilities so far understood to be restricted to humans, such as using collected knowledge to generate causal graphs or identifying background causal context from natural language. As a result, LLMs may be used by human domain experts to save effort in setting up a causal analysis, one of the biggest impediments to the widespread adoption of causal methods. Given that LLMs ignore the actual data, our results also point to a fruitful research direction of developing algorithms that combine LLMs with existing causal techniques. Code and datasets are available at https://github.com/py-why/pywhy-llm.

## 1 Introduction

Recent advances in scaling large language models (LLMs) have led to breakthroughs in AI capabilities. As language models increase in number of parameters and are trained on larger datasets, they gain complex, emergent behaviors, such as abilities to write code in programming languages, generate stories, poems, essays, and other texts, and demonstrate

---

*Authors listed alphabetically.

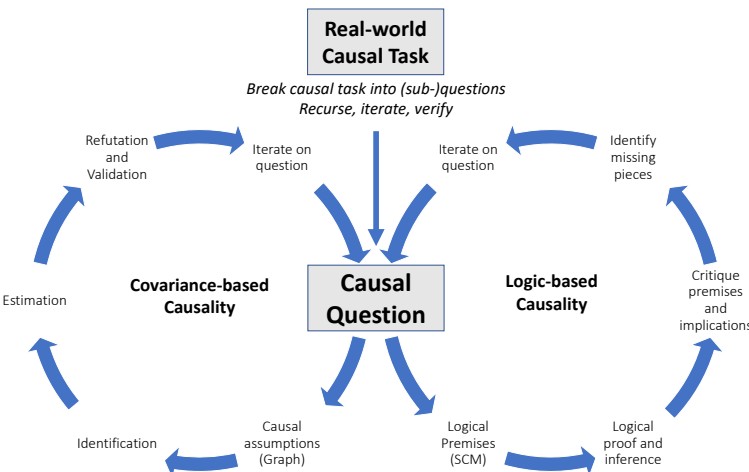

Figure 1: When tackling real-world causal tasks, people strategically alternate between logical- and covariance-based causal inferences as they formulate (sub-)questions, iterate, and verify their premises and implications. Now, LLMs may have the capability to automate or assist with every step of this process and seamlessly transition between covariance- and logic-based causality.

strong performance in certain reasoning tasks (Chen et al., 2021; Nguyen & Nadi, 2022; Bubeck et al., 2023; Katz et al., 2023; Wei et al., 2022a). Impressively, when asked to explain their outputs, update their conclusions given new evidence, and even generate counterfactuals, LLMs can create plausible responses (Nori et al., 2023; Lee et al., 2023a;b). This apparent capacity for both implicit and explicit consideration of causal factors has created excitement towards understanding their reasoning capabilities (Hobbhahn et al., 2022; Kosoy et al., 2022; Willig et al., 2022; Liu et al., 2023; Zhang et al., 2023). Figure 2(a) shows an example of such generation of causal arguments[1].

At the same time, LLMs are imperfect: they can make absurd claims and are often observed to make basic errors of logic and mathematics, much less complex reasoning (Bubeck et al., 2023; Zhong et al., 2023; Ghazal et al., 2017). Figure 2(b) shows an example of an LLM making basic causal mistakes. This has incited a debate about whether LLMs truly perform causal reasoning, or are simply unreliable mimics, regenerating memorized responses in the context of critical analysis and decision-making tasks across many domains, including health, medicine, legal reasoning, education, policy making and business strategy (Bender et al., 2021; Marcus, 2022). Complicating the situation is the variety of distinct formal and informal methods that people ascribe to 'causal reasoning', and the breadth of purposes for which people apply causal reasoning in some form.

In short, what kinds of causal arguments can LLMs generate, how valid are these arguments, and what causal reasoning workflows can this generation support or automate? This paper clarifies this debate, advancing our understanding of LLM models and their causal implications, and proposes a framework for future research at the intersection of LLMs and causality.

We begin with a recognition of the distinct kinds of causal knowledge and reasoning implicated in this debate, including prior knowledge of general and domain-specific causal mechanisms, intuitive and logical methods for causal and counterfactual reasoning, and covariance-based causality, such as building causal DAGs and inferring causal effects (see Figure 1 for an illustration). We present intriguing results in each of areas and then proceed with deeper analyses to better understand the drivers of LLM behavior and the reliability of their causal capabilities in two areas—causal DAG generation and token causality. Our results indicate that LLMs bring significant new capabilities which are complementary to existing causal methods. They do so by capturing the human domain knowledge relevant to the task, which forms an essential part of any causal analysis. As a result, LLMs have the capability of transforming how causal

---

[1]We use the phrase *generate causal arguments* to distinguish LLM generation of the text of a causal statement or argument with causal reasoning itself.

SYSTEM: You are a business advisor and give clear, well-reasoned but brief (5-15 line) advice in response to questions about what people should do.

USER: A toy shop owner in the western USA wants to decide whether the ad they bought in early December is really better than their previous ads. Here is their sales data:
October: $10,200 ; November: $10,000; December: $13,000; January: $10,100
They now want to decide which ad to show in February. Can you help them decide whether the increase in sales in December was due to the ads? Note that the new ad costs $1000 more to produce, so the toy maker is interested in maximizing their profit.

GPT-4: Based on the available data, it is difficult to definitively attribute the increase in December sales to the new ad, as holiday season and other external factors may have influenced sales. To make a more informed decision, I recommend conducting an A/B test in February by dividing the targeted audience into two groups, showing one group the old ad and the other group the new ad. Compare the sales performance of both groups and calculate the net profit (considering the $1000 additional cost for the new ad) to determine which ad is more effective in driving sales and maximizing profit. This will provide a clearer understanding of the ad's impact on sales.

USER: Thanks. In February, the toy shop owner split their audience into two groups to show them two different ads. They showed the old ad to people on the newspaper; and showed the new ad to people online. When people came into the store, the owner would ask them which ad they saw. In February, they found out that they sold $6000 of toys to people who saw the new online ad, and $4500 to people who saw the old newspaper ad. What should the toymaker do in March?

GPT-4: Based on the February results, the new online ad generated $1,500 more in sales than the old newspaper ad. Considering the new ad costs $1,000 more to produce, the net profit increase is $500. In this case, it seems that the new online ad is more effective in driving sales. For March, the toy shop owner should continue using the new online ad, but also consider monitoring the long-term performance of the ad, as well as any potential changes in the target audience's preferences or external factors that may influence sales. Additionally, the owner may want to explore optimizing the new ad further to maximize its cost-effectiveness and reach.

(a) A GPT-4 powered advisor gives sound causal advice

(b) Continuing the conversation, GPT-4 gives a wrong causal answer

Figure 2: **Probing LLM-ability to build causal arguments.** Two example outputs from an LLM (GPT-4). In the first dialog, the LLM discusses causal issues, such as a potential confounder and recommends an A/B experiment to correctly characterize effects and drive the requested decision-making. The second example continues the conversation and requires arguably the same kind of causal awareness of potential confounders (e.g., the population characteristics and even population sizes of the online and newspaper audience are unknown) but the LLM proceeds regardless and provides an incorrect answer.

analysis is done, with the potential to automate or assist in each step of a causal reasoning process, shown by the arrows in Figure 1.

**Summary of results.** Given a set of variables, causal DAG generation is the task of specifying the causal graph characterizing their data-generating process. Starting with a pairwise causal task, we find that LLM-based methods using variable names as input substantially outperform existing algorithms for building causal DAGs from statistical evidence in data (causal discovery algorithms (Peters et al., 2017)), achieving 97% accuracy on the Tübingen benchmark compared to the previous best of 83%. The benchmark includes over hundred causal relationships from a variety of domains, including physics, biology, zoology, cognitive science, epidemiology, and soil science. We repeat the experiment on a more specialized, real-world task related to medical pain diagnosis and obtain similarly high accuracy for GPT 3.5 and 4 models. On the more challenging task of discovering the full graph, we find that LLMs obtain similar accuracy to recent deep learning-based methods. We extend these experiments to pairwise and full graph datasets that were made public after the LLMs' training cutoff date and find similar high accuracies. The errors made by the two kinds of methods—LLMs using variable names and discovery algorithms using the observed data—are not the same, highlighting the potential of combining these methods for greater accuracy. That said, LLMs do make unexpected mistakes; the surprising part is how few such mistakes are, since LLMs only consider the metadata (i.e., variable names) rather than analyzing the data values. To characterize our measurement constructs, we probe whether LLM has been

trained on and memorized the datasets underlying our benchmarks and lay out the implications for interpreting causal benchmarks.

The second major causal task we consider is token causality (Halpern, 2016). Unlike building causal DAGs, which deals with variables and their effect on each other, token causality considers *individual* events and aims to find what caused them. Inferring causes of an event involves simulation of different counterfactual scenarios, but it also involves human judgment to understand the background context and determine which causes need to be considered. For counterfactual reasoning tasks, we find that LLMs like GPT-4 are capable of answering natural language questions. On a benchmark of counterfactual queries spanning basic physics, logic, and common sense, gpt-4 obtains 92% accuracy, 20 points higher than previously reported accuracy. These counterfactual capabilities also help LLMs to isolate the necessary and sufficient causes given any event, one of the key building blocks of token causality methods. GPT-4 obtains over 86% accuracy in identifying the necessary and sufficient causes on a commonly used benchmark of 15 vignettes, as well as a novel benchmark that avoids any memorization concerns. That said, tasks that heavily depend on an understanding of human factors in deciding the cause of an event, e.g., assessing the normality of a particular action, remain challenging for LLMs: GPT-4 obtains 70% on our benchmark task for assessing normality.

**Implications for causality research.** Irrespective of whether LLMs can perform true causal reasoning (as opposed to merely generate well-formed causal arguments that support such reasoning), their empirically observed ability to perform certain causal tasks is strong enough to provide a useful augmentation for aspects of causal reasoning where we currently rely on humans alone. For example, conventional causal DAG generation and causal effect inference rely strongly on prior domain knowledge of potential causal mechanisms in a system. Current best practice is to rely on human domain experts to provide this knowledge, yet correctly capturing domain knowledge in a formal representation suitable for analysis remains a challenge and is often a primary point of weakness for the validity of causal analyses. LLM capabilities now open the possibility of programmatic access to an array of (memorized or inferred) causal mechanisms, capturing general and domain-specific knowledge, and may augment human domain experts by aiding in bootstrapping, critiquing, etc. Other areas where LLMs provide significant benefit include the ability to understand and formalize causal scenarios, generate relevant formal premises based on background knowledge about the world; and ability to identify and correctly frame challenging causal constraints, validations and refutations, such as negative and positive controls, monotonicity relationships, etc. These are all tasks where previously we relied on human experts alone, and now can be partially or entirely automated with human supervision.

That said, LLMs do have unexpected failure modes. In each of the tasks that we studied, LLMs achieve high average accuracies but also make simple, unpredictable mistakes on certain inputs. Further, their accuracy (and consequently robustness) depends substantially on the prompt used, as observed by Long et al. (2023). We provide some basic empirical tests to probe their robustness to specific prompt language and understand the relative contribution of data memorization. However, more research is needed to understand when LLM outputs can be trusted and increase their robustness, either through external tools or other instances of LLMs themselves.

We close with an updated landscape for causal analysis research and practice. Fully characterizing LLMs inherent capacity to build causal arguments and understanding its underlying mechanisms, requires significant research efforts—and until this is accomplished, it is not prudent to trust LLMs alone in critical decision-making tasks and other causal applications. However, current capabilities are sufficiently advanced to be useful in conjunction with existing methods for formal causal reasoning, discovery, and effect inference. We discuss these opportunities, including the possibility that LLMs may provide opportunities for building tighter automated integration between logical and covariance-based approaches to causality.

**Outline of the paper.** Section 2 reviews background in causality and LLMs. Section 3 provides our key results of causal DAG generation on causal discovery benchmarks and the associated implications for causality effect inference research. Section 4 discusses token causality and the performance of LLMs in generating counterfactual arguments, determining necessary and sufficient causes, and causal judgment tasks. Finally, Section 5 discusses how these LLM capabilities provide a way to augment existing causal analyses with domain knowledge and natural language-based reasoning, while also making it possible to combine different kinds of causal methods in a single analysis.

# 2 Background and Preliminaries

Working with natural language, LLMs bring capabilities to causal questions that are complementary to existing approaches. These capabilities span across hitherto disparate fields of causal enquiry, from specifying models of general causal mechanisms to token causality over events, and from statistical effect inference to logic-based reasoning. As a result, we believe that LLMs offer an opportunity to bring together different approaches to causality and build unified pipelines that seamlessly transition between language-based and data-based analyses. Below we review these different approaches or subfields under which causality is studied. We also provide an introduction to LLMs and summarize methods for probing their capabilities.

## 2.1 Many Kinds of Causality

The science of causality is the study of cause and effect relationships, and is a fundamental tool for understanding and reasoning about the world and how it works. Correct causal reasoning is crucial to making correct decisions, building robust systems, and scientific discovery itself. While advances in causal modeling have formalized core concepts of causality, the varying tasks and problems across domains have led different fields to use related but distinct conceptualizations and tools.

Here we describe three orthogonal categories of causal approaches and tasks. The first axis (covariance- and logic-based causality) distinguishes methodological approaches that emphasize data analysis from those that emphasize logical reasoning. The second axis (type vs. token causality) [2] focuses on the setting for a causal question: Are we asking about causal relationships in general, such as for a population, or asking questions about specific causes for a specific event. The third axis, presented in Section 2.2 focus on the causal question or task itself: are we interested in inferring new relationships, characterizing the strength of specific known or hypothesized relationships, attributing blame or reward for some outcome, etc.

**Covariance- and Logic-based Causality:** Many fields, including statistics, biostatistics, and econometrics, place primary emphasis on *covariance-based causal analysis*. This family of methods uses statistical approaches to infer and estimate the strengths of causal relationships from data (Imbens & Rubin, 2015; Hernán & Robins, 2010). Applications include the evaluation of drug efficacy, understanding effects of novel economic policies, and optimizing business decisions. A causal analysis typically starts with a question whose answer is converted to a statistical estimand and then estimated using statistical estimators that work with available data (Pearl, 2009a).

Other domains such as law, forensic investigations, and fault diagnosis, often emphasize *logic-based causality*, which uses logical reasoning and domain knowledge to reason about causal relationships in systems (Hellner, 2000). For example, some notions of legal liability involve establishing proximate cause for an event based on reasoning about counterfactuals and plausible scenarios (Knobe & Shapiro, 2021).

**Type and Token Causality**: *Type causality* encompasses inference on causal relationships between variables, such as causal effect estimation (Peters et al., 2017). In contrast, *token* causality (also called specific causality) refers to inference of the degree to which specific *events* cause other *events* (Halpern, 2016; Hausman, 2005). That is, token causality is concerned with reasoning about specific events and their causes whereas type causality focuses on variables and their average effects. For example, questions concerning medical science such as "does smoking causes lung cancer" or "what are causes of lung cancer" or "how much does smoking increase the risk of lung cancer" are examples of type-causal inference. Scientists are often interested in type causality questions, because these questions help develop theory and make predictions. For example, average causal effect of an intervention tells us something about its effect on a general population and can be used to inform policy.

In contrast, questions under the umbrella of token causality include "Was Fred's smoking habit responsible for his lung cancer? Or was it his exposure to asbestos?", or "What was the reason this machine failed?". These are questions that concern decision-making in specific situations and their answers need not generalize to other situations. Still, the answers are important to inform a specific decision or conclusion that needs to be made, e.g., deciding a legal case or

---

[2]Type causality is sometimes referred to as general causality. Token causality is also known as specific causality or actual causality, though the latter term also refers to the modeling framework for token causality primarily described in (Halpern, 2016).

fixing a machine in a factory. In practice, beyond simple logical reasoning, to determine token causality, people use mental "forward simulations" of processes to predict the potential outcomes of an event, or attribute the underlying events led to an observed outcome (Hegarty, 2004; Jeannerod, 2001).

## 2.2 Different Causal Tasks and their Connection to Kinds of Causality

Causal tasks or questions seek to reason about cause-and-effect relationships in a system, often with the goal of understanding and improving a desired outcome, as other mechanisms and environment change. This broad umbrella encompasses a large array of distinct causal tasks, with its own developed methodologies. Notably, while some of these tasks intersect—they may share common abstractions (e.g., causal graphs) or the result of one task may be fed as an input of another—an ability to solve one task does not imply an ability to solve the others.

*Specifying the causal graph* is the type-causal task of specifying a graph that represents the underlying causal mechanisms that govern a system. *Causal discovery* is an approach to causal graph specification that attempts to reverse engineer the causal graph from covariance in data. (Peters et al., 2017; Glymour et al., 2019).

*Effect inference* is the task of characterizing the strength and shape of a known or hypothesized causal relationship. While it is most commonly characterized as relying on covariance-based methods, effect inference does rely on logical reasoning approaches for determining validity of causal assumptions (Sharma et al., 2021) and identifying approaches to validation and sensitivity analyses (e.g., negative controls (Lipsitch et al., 2010)). Effect inference is primarily focused in type causality scenarios. However, individual treatment effect estimation and counterfactual estimation straddle the realm of token causality.

*Attribution* is the task of determining the cause or causes of a change. Depending on the application domain, approaches to attribution include both covariance-based and logical-based approaches, and are set in both type and token causality settings. For example, determining the likely cause of performance slowdowns in a large-scale Internet service is usually a covariance-based analysis in a type causality setting (Budhathoki et al., 2021); determining the root cause of a specific error in a system execution can be a covariance-based analysis in an token causality setting (Sharma et al., 2022); whereas determining the cause of a fire in an arson investigation may rely purely on logical-reasoning in an token causality setting (Halpern, 2016). *Judgement* tasks (Gerstenberg et al., 2014) extend attribution tasks to questions of reward or blame assignment for outcomes. They usually incorporate additional considerations of morality, normality, and intent of agents (Sloman & Lagnado, 2015).

These are only some of the many causal tasks. Others include *policy optimization*, *decision-making*, *explanation*, *scientific discovery*, and others.

## 2.3 LLMs and Causality

A large language model is a particular kind of machine learning model, built using transformers, a class of deep neural network architectures (Devlin et al., 2019). The primary task of LLMs is next-word completion. Initially, next-word prediction is based primarily on word distribution probabilities. Further training applies additional human feedback to shape the reward function to take into account factors beyond word distribution probability, such as instruction following and safety (Ouyang et al., 2022).

Recent work has explored causal capabilities of LLMs (Willig et al., 2022). For instance, Long et al. (2023) consider simple graphs of 3-4 nodes and test whether LLMs can recover the correct structure. They consider each variable pair (A,B) and ask an LLM to score two competing statements, one implying that A causes B and the other that B causes A. Tu et al. (2023) consider a harder task using a dataset on medical pain diagnosis and find that LLM-based graph inference obtains poor accuracy. On the question of inferring causality from natural language, Hobbhahn et al. (2022) study whether LLMs can understand causal implications embedded in natural language, i.e., given two sentences, whether (the event in) one sentence is the cause of another. In this paper, we extend this line of work and make two contributions. First, we investigate causal graph generation capabilities of LLMs, as an analog to covariance-based causal discovery, over a broad set of complex real-world datasets and explore the robustness of LLM-based graph generation. As one example, in Section 3.1, we revisit the case study on LLM-based graph graph generation from Tu

et al. (2023) and show that with an appropriate prompt, LLMs can achieve substantially higher accuracy (increasing the F1 score for retrieved edges from 0.21 to 0.68). Second, we probe the ability of LLMs to do counterfactual reasoning and infer necessary or sufficient cause based on natural language description of events. We also provide a unifying framework that shows how LLMs can be used to transfer knowledge between covariance-based and logic-based causal methods for a given real-world problem.

## 2.4   Probing LLM behaviors

The primary input-output mechanism exposed by an LLM is a textual prompt as input and a textual response as output. Given black-box access to an LLM, therefore, the main way to understand its causal capabilities is to probe it with different inputs and observe how its output changes. This probing paradigm of building causal arguments, however, has the classic limitation of *construct validity* for the measurements (Smith, 2005). That is, we may be tempted to ascribe a particular causal capability to an LLM if it answers well on a set of questions related to the capability, but the answers may not necessarily be due to the capability; they may be due to other factors such as exploiting some structure in the questions, or in the case of LLMs, memorizing similar questions that it encountered in its web-scale training set (Carlini et al., 2022). Hence we use multiple probing strategies to understand the causal capabilities of LLMs. In addition to the standard practice of testing LLM performance on existing benchmarks, we conduct memorization tests to estimate the extent to which a benchmark dataset may have been memorized, construct novel datasets to avoid memorization concerns, and conduct semantically-equivalent perturbation and other interventions on the input prompts to understand what the LLM pays attention to.

**Benchmark Tests and Question-Answer Evaluation:** The standard approach to evaluating LLM behavior is to ask it questions and evaluate its answers. Our question is the prompt for the LLM; and the LLM's completion of the text is the answer. Most large scale benchmarks and evaluations follow this approach, using purposefully created benchmarks and/or adapting standardized exams written to assess human capability (Ghazal et al., 2017; Nori et al., 2023; Zhong et al., 2023).

Our evaluation approach to characterizing the causal capabilities of LLMs begins with such question-answering evaluations. For each evaluation, we ask a series of questions or causal challenges and score the correctness of the resulting answer. Then, we probe deeper, to better understand the threats to the validity of the question-answer evaluations, and the likely robustness of LLM capabilities.

**Memorization Test[3]:** The primary threat to the validity of benchmark or question-answer style assessments is that the LLM may have directly memorized the benchmark answers. In this case, our questions are likely not testing the LLM's inherent capability to complete a task (unless memorization is the capability we are testing!)

To test whether the LLM has memorized a particular dataset or benchmark, we give the LLM a partial row of data from the dataset and ask it to autocomplete the remainder of the row. For non-tabular datasets, we present a snippet of the file and ask it to auto-complete from that starting point. To encourage the LLM to succeed, we prepend details about the dataset, such as its name, URL, and description, and also provide few-shot examples. The final measurement from the memorization test is the percentage of rows the LLM was able to regenerate correctly. Supplementary E.1 provides additional details on the procedure.

**Redaction Test:** When we see an LLM generate a correct answer to a question, we are still sometimes unsure why. Is the LLM attending to the appropriate aspects of the question, or has it learned to repeat unreliable patterns that may lead to errors in the future. To better understand what aspects of the prompt or question an LLM is attending to, we use redaction and perturbation tests, motivated from explainable AI methods for NLP Sinha et al. (2021); Danilevsky et al. (2020). First, we redact words in the prompt, one by one, and see how the answer changes each time. Changes in the answer indicate the LLM is attending to the redacted word.

---

[3]Our approach to testing for memorization and prior exposure of the LLM follows the method as developed in Bordt et al. (2024).

# 3   Generating causal graphs using LLMs

The task of building a causal graph is a foundational step in a causal inference workflow. Typically, the goal is to obtain a directed graph where the edges denote the presence and direction of causal effect. Such a graph characterizes the underlying data-generating process (DGP) for a dataset and specifies how a change in one variable may (or may not) affect the others. This graph is then used as a base on which downstream analysis relevant to a task is conducted, such as for effect inference, prediction or attribution. Having the correct graph that encodes causal assumptions is critical for ensuring the correctness of any downstream analysis.

Causal discovery attempts to address the challenge of specifying a causal graph by searching for a graph consistent with observed evidence of covariance between the variables in data Peters et al. (2017). The allure of causal discovery is relying on empiricism and avoiding subjective model misspecification errors that could bias the downstream workflow. The challenge, however, is that it is generally not possible to learn the correct graph for a given dataset, given only observational data. The reason is that multiple graph structures are equally likely given the same data distribution, a set of graphs known as the Markov equivalence class (Pearl, 2009c). In the last two decades, two main approaches have been proposed to overcome this limitation. The first is to restrict data-generating process to specific functional forms under which identification of a single graph is possible (Glymour et al., 2019). In some specific settings, such as adding non-gaussian noise to linear data-generating process (Shimizu et al., 2006) or assuming that all functions are non-linear with additive noise (Zhang & Chan, 2006; Zhang & Hyvärinen, 2009), identification is possible. However, there still exist simple settings that are unidentifiable, e.g., a dataset with linear equations and gaussian noise. The second approach is to utilize the power of deep learning to model covariances of all variables jointly and hope that it improves the quality of learnt graphs. However, the identification issue is still not resolved. As a result, recent evaluations of state-of-the-art causal discovery methods on real-world datasets present a sobering picture of their effectiveness (Kaiser & Sipos, 2022; Tu et al., 2019; Huang et al., 2021).

LLMs offer a fresh perspective on the use of statistical algorithms for constructing a causal graph. This perspective focuses on the *metadata* associated with variables in a dataset, rather than their data values. Typically, such metadata-based reasoning is done by human domain experts when they construct causal graphs. By looking at the names of variables and the problem context, for example, people can construct a causal structure based on their knowledge of physics, common sense, or specialized domain knowledge. However, this is a challenging process, hindering the widespread use of such metadata-based reasoning.

We find that LLMs can bridge this gap by filling in the domain knowledge that earlier only humans could provide. In contrast to causal discovery algorithms that use data values of variables, LLMs can infer causal structure by reasoning on *metadata* associated with the variables, for example, the name of the variable and the problem context expressed in natural language. In other words, similar to how domain experts formalize their knowledge in a graph, LLMs can use the knowledge from their training data to infer the edges of the causal graph. To differentiate from the existing covariance-based causal discovery, we call the LLMs capability as *knowledge-based* causal graph generation.

Remarkably, LLMs practicing knowledge-based causal graph generation outperform state-of-the-art covariance-based algorithms on causal discovery benchmarks. Below we present experiments on inferring causal structure using LLMs. We first start with pairwise causal discovery: a simple task involving two variables where the goal is to decide the direction of causal effect, across a range of domains. Next, we will study the problem of full graph discovery in two datasets: one in medicine and one in climate science. While the first task includes many "simple" relationships that an average person is expected to answer correctly, the other two tasks require specialized knowledge (in neuropathic pain and arctic atmosphere science respectively) in order to check whether LLM's graph generation capabilities extend to complex domains.

## 3.1   Pairwise causal edge inference: Inferring causal direction among variable pairs

We start with the pairwise causal edge inference task (Hoyer et al., 2008). Typically, these tasks are set up as discovering causal relations between two variables based on observed data. In this work, instead of relying on observed data, we directly ask large language models whether a variable causes another variable.

| Variable A | Variable B | Domain |
|---|---|---|
| Age of Abalone | Shell weight | Zoology |
| Cement | Compressive strength of concrete | Engineering |
| Alcohol | Mean corpuscular volume | Biology |
| Organic carbon in soil | Clay content in soil | Pedology |
| PPFD (Photosynthetic Photon Flux Density) | Net Ecosystem productivity | Physics |
| Drinking water access | Infant mortality | Epidemiology |
| Ozone concentration | Radiation | Atmospheric Science |
| Contrast of tilted Gabor patches | Accuracy of detection by participants | Cognitive Science |
| Time for 1/6 rotation of a Stirling engine | Heat bath temperature | Engineering |
| Time for passing first segment of a ball track | Time for passing second segment | Basic Physics |

Table 1: Example cause-effect pairs from the Tübingen benchmark. The task is to determine whether Variable A causes Variable B, or vice-versa.

### 3.1.1 Tübingen cause-effect pairs dataset

This dataset (Mooij et al., 2016) consists of data for 108 different cause-effect pairs selected from 37 datasets from various domains, including meteorology, biology, medicine, engineering, and economics (see examples in Table 1). As directionality of an edge is a fundamental building block for learning the full graph, it is a widely used dataset for benchmarking causal discovery algorithms. However, the Markov equivalence class of graphs admissible for a two-node graph (A,B) contains both $A \rightarrow B$ and $B \rightarrow A$. Therefore, the dataset remains a challenging one for causal discovery algorithms, with many recent methods achieving less than $80\%$ accuracy, as shown in Table 2. The best known accuracy is 83%, achieved by the Mosaic algorithm (Wu & Fukumizu, 2020).

We now apply LLMs to the pairwise edge inference problem. We extract the names of each variable from the benchmark and use them to construct prompts for the LLMs. Example prompts are shown in Suppl. A.1 (Table 14). We start with a simple prompt that asks, "Does changing A cause a change in B?", where A and B are the variable names. For a given pair $(A, B)$, we ask the question in both directions and then take the mean accuracy. We choose the mean accuracy since it maps correctly to the accuracy of any method choosing between $A \rightarrow B$ or $B \rightarrow A$. Specifically, the LLM-based method obtains an accuracy of 0 if it answers both questions incorrectly. If it answers one of the questions correctly (indicating that none cause each other or both cause each other), then it effectively chooses the output at random ($A \rightarrow B$ or $B \rightarrow A$). We also report weighted accuracy, as recommended by Mooij et al. (2016) to avoid the effect of overcounting some similar pairs.

**Results.** The second half of Table 2 shows the performance of different LLMs on the task. Similar to prior studies on capabilities of LLMs (Wei et al., 2022a), we observe the emergent behavior that only text-davinci-002 and above achieve a non-trivial accuracy than random chance. With GPT3.5 class models (text-davinci-003 and gpt-3.5-turbo), accuracy of LLM-based method reaches 83% and is competitive to the covariance-based causal discovery algorithms. This suggests that the ability of inferring causal direction emerges as the size of the model is increased. It may also depend on the specific training data and the use of reinforcement learning from human feedback (RLHF) for instruction finetuning.

Following recent work showing the value of the prompt tuning for obtaining high-quality results (Wei et al., 2022b; Long et al., 2023), we employ two different kinds of prompts. In the first, we prepend the prompt with the message, "You are a helpful assistant for causal reasoning", with the intention of steering the LLM's output space towards more causally consistent answers. This simple message of *asking* gpt-turbo-3.5 (i.e., ChatGPT) to be an agent of causal reasoning increases the (weighted) accuracy by almost 5% to 86.9%, consistent with the impact of prompt engineering observed by (Long et al., 2023). The accuracy is higher than previously reported LLM-based accuracy on the task from Choi et al. (2022) that used the davinci-instruct-beta model (83% unweighted accuracy). Moreover, by changing the prompt to instead ask a single question that asks the LLM to output the more likely causal direction between $A \rightarrow B$ or $A \leftarrow B$ while explaining its reasoning in a "step-by-step" manner (see Suppl. A.1), the weighted accuracy improves to

| Model | Acc. | Wt. Acc. |
|---|---|---|
| Slope (Marx & Vreeken, 2017) | 0.75 | 0.83 |
| bQCD (Tagasovska et al., 2020) | 0.68 | 0.75 |
| PNL-MLP (Zhang & Hyvarinen, 2012) | 0.75 | 0.73 |
| Mosaic (Wu & Fukumizu, 2020) | 0.83 | 0.82 |
| ada | 0.50 | 0.50 |
| text-ada-001 | 0.49 | 0.50 |
| babbage | 0.51 | 0.50 |
| text-babbage-001 | 0.50 | 0.50 |
| curie | 0.51 | 0.52 |
| text-curie-001 | 0.50 | 0.50 |
| davinci | 0.48 | 0.47 |
| text-davinci-001 | 0.50 | 0.50 |
| text-davinci-002 | 0.79 | 0.79 |
| text-davinci-003 | 0.82 | 0.83 |
| LMPrior (Choi et al., 2022) | 0.83 | - |
| gpt-3.5-turbo | 0.81 | 0.83 |
| gpt-3.5-turbo (causal agent) | 0.86 | 0.87 |
| gpt-3.5-turbo (single prompt) | 0.89 | 0.92 |
| gpt-4 (single prompt) | **0.96** | **0.97** |

Table 2: Accuracy of different versions of GPT on the Tübingen cause-effect pairs dataset. The best LLM performance outperforms the current state-of-the-art covariance-based approaches that rely on observational data of the two variables. Weighted accuracy weights individual pairs to account for overcounting due to some pairs sharing the same source dataset. The causal agent is gpt-3.5-turbo with system message set as "You are a helpful assistant for causal reasoning.". LMPrior uses davinci-instruct-beta.

92.1%. Finally, using gpt-4 with this prompt, the weighted accuracy shoots up to 97%. This accuracy is substantially higher than that of the best-known covariance-based causal discovery method.

**Understanding LLM output.** The combined single prompt allows us to inspect the LLM output, since it provides an explanation in addition to its final answer. For the examples below, we show the output from gpt-3.5-turbo. Let us look at the example, (age of abalone, length of abalone) where the LLM correctly outputs that changing the age of an abalone is more likely to change its length. Figure 3(a) shows the LLM's output. Its explanation gives the impression that it can *reason* about the problem using its knowledge to arrive at the correct answer:

However, on another pair that arguably should involve similar reasoning, (age of abalone, diameter of abalone), the LLM outputs an incorrect answer and its explanation does not make much sense (Figure 3(b)).

The fact that the LLM can make such errors is not surprising, since it is not explicitly optimizing for being consistent with prior knowledge or causal mechanisms. However, it is remarkable that such errors occur infrequently: The LLM made only 12 errors in the dataset of 108 pairs. On some other pairs where the LLM's output did not match the ground-truth, it reveals possible ambiguity in the variable names, that can be fixed by adding more context to the question. Consider the example of (ozone concentration, radiation) below where the LLM initially provides an incorrect answer.

LLM: 1. Ozone is a gas in Earth's atmosphere that absorbs most of the Sun's ultraviolet (UV) radiation.

2. Chlorofluorocarbons (CFCs), pollutants that were once widely used in refrigerants and aerosol sprays, have been found to destroy ozone molecules in the atmosphere.

3. The destruction of the ozone layer in the upper atmosphere causes more UV radiation to reach the surface of Earth.

SYSTEM: You are a helpful assistant for causal reasoning.

USER: Which cause-and-effect relationship is more likely?
A. changing the age of an abalone causes a change in its length.
B. changing the length of an abalone causes a change in its age.
Let's work this out in a step by step way to be sure that we have the right answer. Then provide your final answer within the tags <Answer>A/B</Answer>.

GPT-3.5-TURBO: 1. Cause: Changing the age of an abalone
Effect: a change in its length.
2. Cause: Changing the length of an abalone
Effect: a change in its age.
Based on the biological knowledge about abalones, it is more likely that changing the age of an abalone causes a change in its length. This is because the length of an abalone is known to increase as it ages, and not the other way around. Therefore, the correct answer is <Answer>A</Answer>.

(a) Correct answer

SYSTEM: You are a helpful assistant for causal reasoning.

USER: Which cause-and-effect relationship is more likely?
A. changing the age of an abalone causes a change in its diameter.
B. changing the diameter of an abalone causes a change in its age.
Let's work this out in a step by step way to be sure that we have the right answer. Then provide your final answer within the tags <Answer>A/B</Answer>.

GPT-3.5-TURBO: Step 1: Determine which variable is the cause and which is the effect.
In Option A, "age" is the cause and "diameter" is the effect.
In Option B, "diameter" is the cause and "age" is the effect.
Step 2: Consider which variable is more likely to cause a change in the other.
Abalone growth is primarily determined by environmental factors such as water temperature, food availability, and water quality. It is unlikely that changing the age of an abalone would cause its diameter to change significantly. On the other hand, changes in the environment could potentially impact the size of the abalone, and therefore, its diameter could change. Therefore, Option B is more likely: changing the diameter of an abalone causes a change in its age.
<Answer>B</Answer>

(b) Incorrect answer

Figure 3: **Probing reasoning of LLMs for deciding causal direction.** Two example outputs from a LLM (gpt-3.5-turbo). The first gives the impression of the LLM performing causal reasoning to arrive at the correct answer—that age may cause the length of abalone to increase. The second example asks a question that requires arguably the same causal knowledge but the LLM proceeds with an incoherent argument and provides an incorrect answer.

4. Increased UV radiation can lead to various negative effects such as skin cancer, cataracts, and damage to crops.

From these facts, we can determine that a decrease in ozone concentration would lead to an increase in radiation. Therefore, Option A is more likely:

<Answer>A</Answer>

The correct answer according to the benchmark is that radiation causes ozone concentration. However, the LLM interpreted the question as about the ozone concentration in the stratospheric ozone layer, whereas the question was about ground-level ozone concentration measured in a city. After adding this context, the LLM's answer changes,

USER: Wait, this question is not about stratospheric ozone concentration. Rather, I'm asking about the ground-level ozone concentration measured in a city and the corresponding ground-level radiation due to sunlight.

GPT-3.5-TURBO:: I see, thank you for clarifying. In that case, the answer is likely option B: changing radiation causes a change in ground-level ozone concentration. This is because ground-level ozone is formed through a chemical reaction between nitrogen oxides (NOx) and volatile organic compounds (VOCs) in the presence of sunlight. This reaction is known as photochemical smog and occurs more frequently on hot, sunny days when there is more radiation. Therefore, an increase in radiation would likely lead to an increase in ground-level ozone concentration.

<Answer>B</Answer>

Finally, in some cases, the direction of the cause-effect relationship is not absolute and depends on other supporting information. For the pair (stock return of Hang Seng bank, stock return of HSBC Hldgs.), the LLM correctly determines that Hang Seng is a subsidiary of HSBC Hldgs. but its answer does not match the benchmark's label. HSBC Hldgs. owns a majority stake in Hang Seng bank, so the benchmark answer indicates that any change in Hang Seng's stock return should also affect HSBC Hldgs. stock return. At the same time, since stock markets do not follow any fixed pattern, it is possible that a major change in HSBC Hldgs. stock return causes people to invest more or less in Hang Seng Bank and hence affect its stock return, as the LLM output argues.

LLM: To determine which cause-and-effect relationship is more likely, we need to consider the nature of the two companies and their relationship with each other.

Hang Seng Bank and HSBC Holdings are two separate companies but they are both major players in the Hong Kong financial market. Hang Seng Bank is a subsidiary of HSBC Holdings and is listed on the Hong Kong Stock Exchange. HSBC Holdings is a multinational banking and financial services company that operates in over 60 countries worldwide.

Given that relationship and the fact that Hang Seng Bank is a subsidiary of HSBC Holdings, it is more likely that:

<Answer>B</Answer>changing the stock return of HSBC Hldgs causes a change in the stock return of Hang Seng Bank.

**Creating a novel Tubingen dataset.** It is possible that LLMs achieve high accuracy in predicting causal direction because Tubingen pairs is a popular dataset which may have been included in LLMs' training data. To test generalization of LLMs to causal pairs that were not a part of LLMs' training data, we build a new dataset. Since the training cutoff date for the models we use (GPT-3.5-turbo and GPT-4) is September 2021, we aim to collect cause-effect pairs that were published after the cutoff date. To do so, we use Google Book search to search for cause-effect pairs in books published on or after Oct 2021. We first categorize pairs from the original dataset into domains such as biology, engineering, medicine, and so on. For each of these categories, we search for appropriate books that describe a causal relationship in their text. Based on this procedure, we obtain a dataset with 67 variable pairs. For instance, the variable pair, "battery capacity", "ambient temperature" was derived from the book, Battery Management System and Its Applications, published in 2022. For a sample list of variables, see Appendix A.3. The full dataset is available at `https://github.com/py-why/pywhy-llm`.

On this dataset, we applied the exact same (single) prompt from the original benchmark. GPT-3.5-turbo and GPT-4 obtain 80.3% and 98.5% accuracy respectively, indicating that the capability to identify causal direction generalizes to variable pairs outside of popular datasets.

### 3.1.2 Neuropathic pain dataset

We now consider a ground-truth causal graph constructed by medical experts on neuropathic pain diagnosis (Tu et al., 2019). It contains relationships between different nerves and the associated symptoms that patients express. Table 3 shows example of edges in the graph. Unlike the Tübingen benchmark which contains some commonly known variables, understanding the names of variables in this graph requires specialized medical knowledge.

We evaluate LLMs on a pairwise causality task similar to the Tubingen benchmark. We do so by creating a pair for each edge in the graph and setup the task to determine the direction of the edge. That is, for all the edges present in the ground-truth graph, we ask an LLM to infer the correct direction of causality. We obtain a dataset of 475 edges.

Table 4 shows the accuracy of different LLMs in determining the direction of edges. Similar to the Tübingen benchmark, smaller language models are unable to obtain accuracy substantially higher than random chance (50%). In addition, even text-davinci-003, the largest text completion model in our comparison set, is unable to obtain a high accuracy using the two-sided prompt. However, chat-based models, trained using human feedback, are able to distinguish causal

| Variable A | Variable B | Dir. |
|---|---|---|
| Right L1 Radiculopathy | Right adductor tendonitis | → |
| Pharyngeal discomfort | Right C3 Radiculopahty | ← |
| Right L5 Radiculopathy | Lumbago | → |
| Left PTA | Left L4 Radiculopahty | ← |
| Left T3 Radiculopahty | Toracal dysfunction | → |
| DLS L5-S1 | Right S1 Radiculopathy | → |
| Left C3 Radiculopathy | DLS C2-C3 | ← |
| Left C7 Radiculopathy | Left medial elbow problem | → |
| Right Ischias | Right L5 Radiculopathy | ← |
| Right Morton trouble | Right L5 Radiculopathy | ← |

Table 3: Example cause-effect pairs from the Neuropathic pain diagnosis benchmark. 'Dir.' refers to the ground-truth causal direction between the variables.

| Model | Accuracy |
|---|---|
| ada | 40.1 |
| text-ada-001 | 50.0 |
| babbage | 50.0 |
| text-babbage-001 | 50.9 |
| curie | 50.0 |
| text-curie-001 | 50.0 |
| davinci | 38.4 |
| text-davinci-001 | 50.0 |
| text-davinci-002 | 51.7 |
| text-davinci-003 | 55.1 |
| gpt-3.5-turbo | 71.1 |
| gpt-3.5-turbo (neuropathic pain expert) | 75.1 |
| gpt-4 | 78.4 |
| gpt-4 (neuropathic pain expert) | 84.3 |
| text-davinci-003 (single prompt) | 86.0 |
| gpt-3.5-turbo (single prompt) | 85.5 |
| gpt-4 (single prompt) | **96.2** |

Table 4: Accuracy of different versions of GPT on the inferring the edge directions of the Neuropathic pain diagnosis graph. As with the Tübingen dataset, LLMs like gpt-3.5-turbo obtain more than 85% accuracy on determining the direction of edges. The causal agent is gpt-3.5-turbo with a system message set as "You are a helpful assistant for causal reasoning."

direction accurately. With the two-sided prompt, gpt-3.5-turbo achieves the highest accuracy of 75%. The choice of prompt also has a big impact. As with the Tübingen dataset, using single prompt per variable pair leads to a significant increase in accuracy: text-davinci-003 now matches gpt-3.5-turbo and both models obtain more than 85% accuracy. While the performance of these models is impressive, gpt-4 obtains an even higher accuracy (96%), yielding more than 457 correct responses out of 475.

**Understanding LLM output.** We use the output of gpt-3.5-turbo (single prompt) to understand how LLMs obtain such high accuracies. On manual inspection of the output by gpt-3.5-turbo, we find that it correctly understand the medical terms in almost all cases. In the example below, the LLM provides the correct definitions of DLS T5-T6 and Left T6 Radiculopathy (without access to any such information in the prompt) and provides the correct causal direction. Suppl. A.2 provides the LLM output for this pair. However, even as it provides the correct definitions, it can provide erroneous output in unpredictable ways. Here it went from the correct definition of the same concept (DLS) to confusing it with the "T5-T6 region" associated with DLS T5-T6 (see Suppl. A.2 for the full output).

Moreover, the example in Table 5 adds evidence that LLMs may not be capable of consistent and coherent reasoning, even as they output correct answers for a majority of the causal questions. In the left panel, the LLM provides the correct reasoning for most of its answer but ultimately chooses a final answer inconsistent with its own reasoning (A instead of B). Unlike reasoning errors, fortunately self-consistency is an easier problem to verify using the LLM itself. We find that gpt3.5-turbo and earlier class of models are unable to check consistency of an output, but GPT-4 is able to. Therefore, it may be possible to pass all LLM outputs through a GPT-4 verifier to ensure self-consistency of the outputs. The right panel of Table 5 shows the verifier prompt and output from GPT-4 (the question and answer from gpt-3.5-turbo are not fully shown for brevity).

| Initial response | Critiquing the response |
|---|---|
| SYSTEM: You are an expert on neuropathic pain diagnosis.
USER: You are a helpful assistant to a neuropathic pain diagnosis expert. Which cause-and-effect relationship is more likely?
A. Right C7 Radiculopathy causes DLS C6-C7.
B. DLS C6-C7 causes Right C7 Radiculopathy.
Let's work this out in a step by step way to be sure that we have the right answer. Then provide your final answer within the tags <Answer>A/B</Answer>.
GPT-3.5-TURBO: *First, we need to understand what each term means. DLS stands for degenerative lumbar spondylosis, which is a condition where the spinal discs in the lower back degenerate and cause pain. C6-C7 refers to the sixth and seventh vertebrae in the cervical (neck) spine. Radiculopathy is a condition where a nerve root is compressed or damaged, causing pain, numbness, or weakness in the corresponding area of the body.*

*Based on this information, it is more likely that option A is the correct cause-and-effect relationship. This is because radiculopathy is often caused by compression or damage to a nerve root, which could be the result of degenerative changes in the spine (such as DLS). In this case, the degeneration in the lower back (DLS) could be causing compression or damage to the nerve root at the C7 level, resulting in right C7 radiculopathy.*

*Therefore, the answer is <Answer>A</Answer>.* | USER: Analyze the output from an AI assistant. Is the final answer (A) consistent with the reasoning provided by the assistant?

Question:
{{USERPROMPT-FROM-BEFORE}}

AI assistant: {{ANSWER-FROM-BEFORE}}

GPT-4-WEB: *The final answer (A) is not consistent with the reasoning provided by the assistant. The reasoning suggests that DLS C6-C7 could cause Right C7 Radiculopathy, which corresponds to option B. The correct answer should be <Answer>B</Answer>.* |

Table 5: **Critiquing LLM output using another LLM instance.** To increase robustness of an LLM's response, we can use GPT-4 as a critic. The left panel shows an incorrect reply from gpt-3.5-turbo wherein the reasoning is correct but the LLM outputs the incorrect option (A). We create a special "critique" prompt that asks gpt-4 to evaluate the response from an AI assistant for self-consistency. gpt-4 finds the logical inconsistency and provides the correct answer.

In addition to self-consistency, when LLM outputs do not match the ground-truth, they can sometimes help critique the ground-truth itself. For instance, the ground-truth graph contains an edge from Left L5 Radiculopathy to Obesity. However, the LLM output provides the opposite direction: obesity can cause L5 Radiculopathy, which is corroborated by research that claims obesity as a risk factor for radiculopathy (Atchison & Vincent, 2012; Tomić et al., 2009).

LLM: Step 1: Understand the terms

- Neuropathic pain: pain caused by damage or disease affecting the nervous system

- L5 Radiculopathy: a condition where the nerve root that exits the spine at the L5 level is compressed or irritated, causing pain, numbness, or weakness in the leg

- Obesity: a medical condition characterized by excessive body fat that increases the risk of health problems

Step 2: Analyze the cause-and-effect relationship
- Option A: Left L5 Radiculopathy causes Left Obesity. This relationship does not make sense because L5 radiculopathy is a nerve condition that affects the leg, while obesity is a whole-body condition related to excessive body fat. It is unlikely that a leg nerve condition would cause obesity.

- Option B: Left Obesity causes Left L5 Radiculopathy. This relationship is more plausible because obesity can put extra pressure on the spine and nerves, leading to nerve compression or irritation. It is possible that obesity could cause L5 radiculopathy.

Step 3: Choose the more likely cause-and-effect relationship

- Based on the analysis, Option B is more likely: Left Obesity causes Left L5 Radiculopathy.

<Answer>B</Answer>

### 3.1.3 Other properties of pair-wise relationships

In addition to asking the LLM about the existence and direction of a causal relationship between a pair of edges, we can also ask many other questions as well. For example, we can ask if a relationship may be confounded by other variables, or what variables may mediate or moderate the relationship. We can ask whether a causal relationship is monotonic, whether its effects are homogeneous or not with respect to other properties. Other useful properties to identify include the time-lag of a causal relationship, whether it is stable, and whether it is subject to spillover effects. Designing prompts for these questions should be a straightforward extension of the pairwise edge existence prompts.

### 3.2 Full graph generation

We now extend the pairwise analysis to generating the full graph. Given a set of variables, a straightforward extension is to construct a list of all possible variable pairs and repeat the pairwise test for each pair. However, full graph generation provides additional challenges. First, the pairwise analysis assumed the existence of an edge and our goal was simply to determine its direction. In the graph generation problem, there are three possible options for each variable pair: $A \rightarrow B$, $A \leftarrow B$, or no edge exists. Second, graph generation requires the ability to distinguish between direct and indirect causes, given the other nodes in the graph. For example, if the true relationship is $A \rightarrow B \rightarrow C$, then it may be correct to output both $A \rightarrow B$ and $A \rightarrow C$ in a pairwise task, but for graph generation, outputting $A \rightarrow C$ is an incorrect answer. Moreover, the decision depends on which variables are present in the input variable set for the graph generation task. For example, if $B$ is not one of the variables in the input variable set, then $A \rightarrow C$ is a valid edge and should be included.

Due to these reasons, it is unclear how well a simple extension of the pairwise task using LLMs will transfer to the full graph generation problem. For simple graphs over 3-4 variables, Long et al. (2023) show that LLMs can obtain promising accuracy in inferring the edges. Below we report the results of applying the pairwise LLM prompts on more complex, real-world datasets involving specialized knowledge: on pain diagnosis and on artic sea ice coverage.

| Model | Precision | Recall | F1 |
|---|---|---|---|
| Random | 0.25 | 0.5 | 0.33 |
| chatGPT (Tu et al., 2023) | 1 | 0.12 | 0.21 |
| text-davinci-003 | 0.59 | 0.68 | 0.63 |
| gpt-3.5-turbo (single prompt) | 0.66 | **0.71** | **0.68** |
| gpt-4 (single prompt) | **0.74** | 0.58 | 0.65 |

Table 6: Accuracy of different versions of GPT on the Neuropathic pain dataset.

### 3.2.1 Neuropathic pain dataset

We continue our exploration with the neuropathic pain dataset. It has 221 variables, so the total number of possible variable pairs are $C(221, 2) = 24310$. Before conducting full graph generation, we utilize a 100-pair dataset provided by the authors of the dataset to evaluate LLM's capability to infer causal relationships (Tu et al., 2023). It has 50 variable pairs that form edges in the graph and 50 pairs that do not form an edge. We notice that many of the variable names are in Swedish, so we employ an LLM (gpt-4-web) to translate the names to English as a preprocessing step.

**Results.** Table 6 shows the results of different LLM-based methods. As a baseline, using an algorithm that return either direction at random for each pair, we would obtain 0.5 recall and 0.25 precision on recovering the true edges, leading to an F1 score of 0.33. Using a simple prompt, *"A causes B. R and L refer to the right and left sides of the body respectively. Answer with true or false."*, Tu et al. (2023) use chatgpt-3.5-web to obtain an F1 score of 0.21. This LLM performance is worse than the random baseline which may indicate that LLMs are not useful for full graph generation. However, with a different prompt, we find that the same model can provide over 3X increase in the F1 score. We provide the LLMs with an identical prompt to the "single" prompt from the pairwise task, with one addition: we add a third option, "C: No causal relationship exists". With this prompt, gpt-3.5-turbo (API analogue of chatgpt-3.5-web) obtains 0.68 F1 score. This score is double the F1 score expected from a random guess baseline, indicating that the output from LLMs provide a non-trivial utility for inferring causal edges.

**Using a novel medical dataset.** It is possible that the Neuropathic dataset may be a part of the training set for the LLMs. Therefore, we consider a causal DAG proposed by Mascaro et al. (2023) that was published online in February 2022[4], a few months after the training cutoff date of the LLMs considered. The DAG captures expert-provided relationships concering COVID-19 disease processes. We consider a subset of the full DAG containing 11 nodes, as reported by Mascaro et al. in their paper. Variables correspond to concepts in respiratory medicine, such as "Alveolar Epithelial infection" and "Viremia".

We apply the same prompt that we used for the Neuropathic dataset. GPT-4 obtains an F1 score of 0.73 which is comparable to the best-reported F1 on the Neuropathic dataset (F1=0.68).

### 3.2.2 Arctic sea ice dataset

To evaluate LLM's ability to generate causal graphs, we now consider a dataset from a different field of science: arctic sea ice and atmospheric science. This dataset is on the drivers of arctic sea ice thickness (or coverage): what causes the arctic sea coverage to increase or decrease? In a recent publication, Huang et al. (2021) provide a domain knowledge-based graph with 12 variables and 48 edges. Variables in the graph include total cloud water path, sea level pressure, geopotential height, meridional and zonal wind at 10m, net shortwave and longwave flux at the surface. Importantly, the graph contains some double-sided edges.

Since the number of variables is low (12), we are able to conduct an experiment to generate the full graph using LLMs. Huang et al. (2021) evaluate the accuracy of three recent discovery algorithms: temporal causality discovery framework (Nauta et al., 2019), NOTEARS (Zheng et al., 2018), and DAG-GNN (Yu et al., 2019). They evaluate the algorithms on the normalized Hamming distance (NHD) between the predicted graph $G'$ and the ground-truth graph $G$.

---

[4]The first article describing the dataset was published on medRxiv in February 2022 and the dataset was released in the same month (URL: `https://osf.io/bynr6/`). The OSF project, itself, was created on 1st October 2021.

| Algorithm | NHD | No. of predicted edges | Baseline NHD | Ratio |
|---|---|---|---|---|
| TCDF | 0.33 | 9 | 0.39 | 0.84 |
| NOTEARS (Static) | 0.33 | 15 | 0.44 | 0.75 |
| NOTEARS (Temporal) | 0.35 | 7 | 0.38 | 0.92 |
| DAG-GNN (Static) | 0.32 | 23 | 0.49 | 0.65 |
| DAG-GNN (Temporal) | 0.34 | 16 | 0.44 | 0.77 |
| gpt-3.5-turbo | 0.33 | 62 | 0.76 | 0.43 |
| gpt-4 | **0.22** | 46 | 0.65 | **0.34** |

Table 7: Normalized hamming distance (NHD) for different causal discovery algorithms. Since NHD depends on the number of predicted edges, we compare the ratio of NHD and baseline NHD across algorithms. A lower NHD ratio is better. LLM-based graph generation (gpt-3.5-turbo) obtains comparable NHD and the lowest NHD ratio compared to recent covariance-based discovery algorithms.

For a graph with $m$ nodes, the distance is given by $\sum_{i,j=1}^{m} \frac{1}{m^2} 1_{G_{ij} \neq G'_{ij}}$, the number of edges that are present in one graph but not the other, divided by the total number of all possible edges. The distance is zero if $G'$ outputs exactly the same edges as $G$. For reference, since the number of nodes and edges in our ground-truth graph are 12 and 48 respectively, if $G'$ outputs 48 completely different edges compared to the 48 edges of $G$, the NHD between $G$ and $G'$ will be 0.66. However, if we output no edges at all, the NHD will be better than the algorithm, 0.33. Since the NHD depends on the number of edges returned by a discovery algorithm, we compare the ratio of NHD of the discovery algorithm and a "floor" baseline that outputs the same number of edges but all of them are incorrect. A lower ratio implies the multiple by which the graph discovery algorithm is better than the worst baseline returning the same number of edges.

Table 7 shows the NHD metric for the three covariance-based causal discovery algorithms and knowledge-based graph generation with LLMs. For each algorithm, we also show the number of predicted edges and the NHD for a worst case algorithm that predicts the same number of edges. NHD for the three covariance-based algorithms is similar. However, the ratio of NHD for the algorithm and the baseline algorithm provides the relative improvement that the algorithm provides compared to the baseline. Using the ratio metric, we see that DAG-GNN (static) performs the best and has the lowest Ratio distance.

For LLM-based graph generation, we use the same prompt as in the neuropathic pain dataset. Using gpt-3.5-turbo, we obtain a normalized hamming distance of 0.33, comparable to the three recently proposed covariance-based causal discovery algorithms. However, the LLM returns 62 edges, so its NHD ratio (0.43) is substantially better than covariance-based algorithms. Finally, gpt-4 obtains the lowest NHD among all algorithms and significantly outperforms state-of-the-art covariance-based discovery algorithms. It outputs almost the same edges as the ground-truth (46) with an NHD of 0.22, one-third less than other algorithms. Its NHD ratio is also the lowest. gpt-4 recovers over half (29) of the edges correctly with a precision of 0.63 (F1 score=0.57), indicating the competitiveness of LLM-based algorithms for causal generation.

**Measuring LLM's NHD metric on a novel dataset.** In addition to the Arctic sea ice dataset, we consider a dataset that was created in 2023 by consultation with medical experts on Alzheimer's disease (Abdulaal et al., 2023). Since the causal graph was created after the training cutoff date of September 2021, it cannot be present in the training data for LLMs. Abdulaal et al. (2023) present an expert-validated causal graph of 11 nodes. Variables in the graph include age, ventricular volume, brain MRI, etc. They use the same prompt that we use for the Arctic sea ice dataset and report NHD of the obtained graphs using different LLMs. Table 16 in Appendix A.3 reports their results on the dataset. The results show that the graph generated by gpt-4 obtains the lowest NHD (0.14) compared to existing existing data-based methods such as NOTEARS (0.22) and DAG-GNN (0.37). Its NHD ratio is also the lowest (0.28). These results, similar to those obtained for the Arctic sea dataset, indicate the generalizability of LLMs like GPT-4 to for inferring full graph over novel datasets. Note that novel datasets may still contain concepts that existed before the LLM's training cutoff date. See Section 5.1 for a discussion on utility of LLMs with respect to novel scenarios.

Which cause-and-effect relationship is more likely? A. changing SLOT1 causes a change in SLOT2. B. changing SLOT3 causes a change in SLOT4. Let's work this out in a step by step way to be sure that we have the right answer. Then provide your final answer within the tags <Answer>A/B</Answer>.

Figure 4: We probe the importance of individual words for getting a correct answer by redacting a random word from a question. We highlight words based on their importance for getting a correct result. A white background indicates that redacting the word did not reduce accuracy. A dark blue highlight indicates that redacting the word reduced accuracy the most.

## 3.3 Probing LLM behavior further

To better characterize the validity and robustness of our experiments at the intersection of LLMs and building causal graphs, we report the results of our LLM probing experiments, described in Sec 2.4.

**Memorization Test Results:** We provide the LLM with the first 3 columns of the dataset (the row ID and 2 variable names), and ask the model to complete the remaining 2 columns of the row (the source dataset name and the ground truth causal direction. We find that GPT-3.5 is able to correctly recall 58% of the remaining cells in the dataset, and recall 19% of the rows without error. Our memorization test with GPT-4 is able to recall 61% of remaining cells in the dataset and 25% of rows without error. These results indicate that the Tübingen dataset is certainly in GPT's training set and that GPT has had an opportunity to memorize the dataset. However, the gap between percentage of inputs memorized and LLMs accuracy is still significant and more work is needed to understand the role of memorization for the inputs that are not captured by our memorization test.

As the neuropathic pain and the arctic sea ice coverage datasets are not available online in a tabular file format, we adapted our memorization test to ask for completions of randomized samples of JSON and code files [5]. We provide the LLM with a contiguous section of text from a randomly chosen location within a file, and ask the model to complete the file from that point. Counting the number of characters that are correctly reproduced by the LLM, provides an indication of whether the dataset has been memorized by the model. Our results, detailed in Section E.1, indicate that the neuropathic dataset has been at least partially memorized by GPT-3.5 Turbo, but not GPT-4, and the arctic sea ice dataset has not been memorized by either GPT-3-5 Turbo or GPT-4.

More generally, we expect knowledge-based causal graph generation to be founded partly on memorized information, but also dependent on the LLM's ability to process and transform seen causal relationships for use in multiple contexts. We can conceptually model the LLM's end-to-end efficacy $P(Y|D)P(D)$, where $P(D)$ represents the likelihood of a causal relationship being memorized by the LLM, and $P(Y|D)$ the likelihood that the relationships can be correctly transformed. Knowing that our benchmark dataset is memorized implies that our experiments can only measuring $P(Y|D)$, but not measure $P(D)$. That is, when evaluating an LLM on an existing benchmark, its performance on the benchmark is a test of how well the LLM's ability to process and transform its knowledge into the necessary causal relationship assuming that it has been trained using some representation of the underlying information.

**Redaction Test:** To characterize whether or not the LLM is attending to the correct words and concepts in a question when answering our pair-wise causal edge inference questions, we use a redaction test. Our redaction test shows what words are important for correctly labeling a causal relationship between two features. Figure 4. shows our results for experiments in gpt-3.5-turbo, averaged across 357 random redaction probes over our Tübingen experiment. If the LLM is correctly reading the question, we expect that it will pay a lot of attention to key instructions and to the options themselves. Here we see that some of the key instruction words that indicate we are asking about causal relationships, such as 'changing', and 'causes' as well as instructions for how to format the final answer in tags are

---

[5]For the neuropathic dataset, we use the JSON representation available on the public web at https://observablehq.com/@turuibo/the-complete-causal-graph-of-neuropathic-pain-diagnosis, and for the arctic sea ice dataset, we use a python file that embeds a 12x12 binary matrix representation of the causal graph, https://github.com/big-data-lab-umbc/cybertraining/blob/2179fb771dcb69d4f302f693b30e057c366f6be1/year-3-projects/team-6/NOTEARS/compute_dist_matrix_between_hyperparam_STATIC_AND_TEMPORAL.py. We omit tests for novel datasets created after LLM training cut off dates.

critically important, each affecting the accuracy of the result in the most redactions. The SLOT phrases are important but redundant (SLOT1 and SLOT4 are identical and SLOT2 and SLOT3 are identical). Therefore the LLM seems more robust to their individual redactions in this test, each individual redaction affecting the accuracy of the result, but less often. While most unimportant words do not affect the accuracy of the result much, we do see the redaction of number of 'minor' words having a surprisingly strong effect, possibly indicating a general sensitivity of the LLM to grammatical correctness.

## 4   LLMs for Token Causality and Causal Judgments

In the previous section, we saw how LLM's ability to capture common sense and domain knowledge allows for a knowledge-based approach to building a causal graph. In this section, we study how LLMs' can bring these capabilities to enable a systematized approach to token causality challenges. Token causality is motivated by problems of attribution and assigning responsibility in real world applications, such as legal reasoning, machine failure debugging, and root-cause analysis for system regressions. For example in tort law, the core problem of deciding how much blame for a plaintiff's injury is attributed to a defendant's action relative to other potentially mitigating events is fundamentally an token causal inference task.

Causal inference researchers have attempted to use formal causal models to define token causality in a way that is consistent with how humans naturally attribute cause and related concepts of responsibility, blame, and explanations to events and their outcomes (Halpern, 2016; Kueffner, 2021). This task has proven exceedingly difficult, because human judgments of causality depend on elements of background context that are difficult to formalize in an SCM. Types of background context that are fundamental to human causal judgments but difficult to formalize in an SCM include:

- **Causal frame**: The set of candidate causal events that are relevant to a particular outcome event. SCMs require relevant causes be included as endogenous variables, but in practice humans rely on domain knowledge and common sense to set the causal frame after the outcome occurs (Icard & Goodman, 2015). For example, there are a multitude of potential causes of forest fires, but in the case of a particular forest fire, upon learning lightning struck and that a hiker was carelessly smoking a cigarette during fire season, we know these are relevant causal events. We would also know whether to ignore "drought" or "lack of controlled burn" if they are not relevant in the case of this particular fire.

- **Necessary causality**: Whether a candidate cause needed to happen for the outcome to occur.

- **Sufficient causality**: Whether a candidate cause's occurance would have led to the outcome event if other causal events had occurred differently.

- **Normality**: The degree to which causal events align with statistical norms or prescriptive norms (social, moral, or legal norms) (Phillips et al., 2015; Kominsky et al., 2015). When agents violate norms, they are typically judged to be more of a cause of resulting outcomes (Phillips et al., 2015; Kominsky et al., 2015). Human causal judgments depend highly on whether candidate cause is a norm violation, or whether it is more or less normal than other causal events.

- **Other human factors**: Other human factors include bias towards action, handling intention and epistemic state, and how bad outcomes are interpreted. When the candidate cause is an agent's behavior, humans tend to ascribe more causality actions (e.g., tossing a cigarette) than to lack of actions (e.g, not doing a controlled burn) (Henne et al., 2017). In addition, when the candidate cause is an agent's behavior, whether the agent acted with intention and knew what they were doing (Nadelhoffer, 2006; Knobe, 2003) matters. Finally, human causal judgments also depend on whether the outcome is undesirable (e.g., causing a forest fire vs. causing a reforestation initiative) (Kominsky et al., 2015).

Sidestepping the challenge of formalizing these concepts into a causal model, LLMs offer an opportunity to capture the necessary and relevant background context for an event directly from natural language description of an event.

| Premise | Counterfactual Question | Multiple-choices answers |
|---|---|---|
| A woman does not order Chinese food. | What would have happened if she had ordered Chinese food? | The woman would have become less hungry.;The woman would have become very hungry.;That is not possible. |
| A woman sees a fire. | What would have happened if the woman had touched the fire? | She would have been burned.;She would not have been burned.;That is not possible.;She would have seen fire. |
| A bird lands in a forest. | What would have happened if a plane had landed in the forest? | The plane would have crashed.;Everything would have been fine.;The plane would have landed safe and sound.;In a forest you will find lots of planes. |
| A plant grows in a planter. | What would have happened if the planter grows in the plant? | That is not possible.;It would have grown quicker.;The plant would have suffered.;The planter would have cultivated the plant. |
| A mortician prepares a corpse. | What would have happened if the mortician had prepared a dinner? | He would have had a delicious dish.;Morticians cannot prepare dinners.;The dinner would have been buried.;The mortician would have killed the corpse. |
| An oil tanker sails across an ocean. | What would have happened if the oil tanker had broken up in an ocean? | There would have been environmental pollution.;That is not possible.;The oil tanker would have continued to carry oil.;The oil tanker would have been saved.; |
| A car crashes into a tree. | What would have happened if the car had parked beneath the tree? | Nothing special would have happe ned.;The car would have been hit by the tree.;That is not possible.;I think it would have crashed into the tree. |
| A child draws a picture. | What would have happened if the child had erased the picture? | The picture would not have been visible.;The picture would have been visible.;That is not possible. |
| A craftsman builds a house. | What would have happened if the house had built a craftsman? | That is not possible.;The house would have been built faster.;Everything would have been fine.;The craftsman would have hands. |
| A doctor washes their hands at work. | What would have happened if the doctor hadn't washed their hands? | The patients could get an infection.;The patients could get better.;That is not possible. |

Table 8: Example scenarios from the CRASS counterfactual reasoning benchmark. The task is to select the best answer choice for the counterfactual question, given a premise.

Given that an LLM is trained on text narratives written by humans, the subjective elements of causal judgments may be encoded as part of the LLM's internal representations.

In this section, we first review the concept of counterfactual reasoning and show that LLMs can answer counterfactual questions about an event with accuracy close to that of humans on a benchmark task. We then present an example scenario to illustrate the challenges of modeling real-world events using token causality and introduce the two key counterfactual concepts used in formal models of token causality: necessity and sufficiency. Using 15 different vignettes commonly used in token causality literature, we provide evidence that LLM can capture the necessary background context and reason about necessity and sufficiency of token causes. Finally, we show how this capability transfers to a causal judgment benchmark task from Big Bench (Suzgun et al., 2022), specifically designed to test LLMs on their ability to infer the normality of a candidate causes.

## 4.1 Building block of token causality: Counterfactual reasoning

Counterfactual reasoning is the process of considering hypothetical situations or alternate realities by altering specific elements or conditions of an actual event or situation (Kahneman & Miller, 1986; Byrne, 2005). Such reasoning is a key element of token causality. To determine the causes of an event, it is important to simulate alternative worlds where an event may not have happened and reason about the consequences. For example, a naive approach to token causality is to use a definition based on *counterfactual dependence*: an event A is a cause of another event B if B would not have happened without A.

Independent of token causality, counterfactual argument making is a desirable skill for a LLM, as it would assist users in decision-making, planning, and offer insights that may not explicitly apparent from the original context. With this motivation, Frohberg & Binder (2022) provide a benchmark called CRASS (Counterfactual Reasoning Assessment) for evaluating the ability of LLMs to answer counterfactual conditional questions, a subset of which is included in the

| Model | Accuracy |
|---|---|
| GPT-3 (Frohberg & Binder, 2022) | 58.39 |
| T0pp (Sanh et al., 2021) | 72.63 |
| text-davinci-003 | 83.94 |
| gpt-3.5-turbo | 87.95 |
| gpt-4 | **92.44** |
| Human annotators | 98.18 |

Table 9: Accuracy of different LLMs on the CRASS counterfactual reasoning benchmark. gpt-4 achieves 92% accuracy, significantly higher than the previous reported accuracy on this benchmark and within six percentage points of human annotators' accuracy.

BIGBench collection of datasets (Srivastava et al., 2022).[6] The benchmark contains 275 instances, where each instance has the LLM select from multiple choice answers to a counterfactual conditional question such as *"A woman opens a treasure chest. What would have happened if the woman had not opened the treasure chest?"*. Table 8 lists some instances from the dataset. Our memorization test indicates the CRASS dataset has not been memorized (see Section E).

The authors report GPT test-davinci-003 as the best performing (zero-shot) LLM on this task, with a top-1 accuracy of 58.39%. Another language model (T0pp) (Sanh et al., 2021), which was finetuned on a multi-task setup of classification tasks, obtains 72% accuracy. This is contrasted with 98.18% accuracy human baseline, indicating that counterfactual predictions remain a challenging task for LLMs.

We revisit the claim using LLMs released after the benchmark's publication. As shown in Table 8, each problem instance has a premise, counterfactual question and a list of answer options. We construct a prompt for the LLM by concatenating the premise and the counterfactual question, and then presenting the answer options as A, B, C, D (for an example prompt, see Suppl. B.1). For gpt-3.5-turbo and gpt-4, we additionally provided a system message, "You are a helpful assistant for counterfactual reasoning."

Table 9 shows the accuracy of different LMs. Remarkably, GPT 3.5 version models show substantial improvement in accuracy over GPT 3. gpt-3.5-turbo obtains an accuracy of 87.95. gpt-4 improves it further to 92.44%, which is 20 percentage points higher than the previous best accuracy. Comparing to the human accuracy on this task, gpt-4 is within six percentage points of average human accuracy. The results indicate that large language models of GPT-3.5 and 4 series represent a substantial jump in LLM ability to generate counterfactual arguments.

**Understanding LLM output.** We now try to analyze some outputs from gpt-4 to understand its counterfactual argument-making capabilities. For full LLM outputs of the prompt discussed here, see Suppl. B.1.As the high accuracy indicates, gpt-4 is able to simulate different scenarios based on the text prompt and answer what-if scenarios. In many cases, when gpt-4's answer does not match the benchmark, it reveals ambiguity in the text used to describe the scenario. In one of the cases, the LLM is asked what will happen if a man catches a water balloon, given that he has caught a ball. The benchmark answer is that he will get wet whereas the LLM output correctly argues that he may get wet or not, depending on whether the balloon bursts as he catches it. Similarly, another question asks what will happen if a man gets very nervous. The benchmark answer indicates that he will pass out, but the LLM output correctly mentions that, *"Passing out due to extreme nervousness is possible, but not guaranteed.* and concludes that the man would not pass out. We suspect that given enough clarifying context, the LLM should be able to correctly answer such questions.

That said, there are a few instances where the LLM does not capture context that people may easily determine through common sense. For instance, one of the questions asks what would happen if a man walks on a bed instead of walking on the street. "Walking on a bed" conveys that the man is likely indoors, but the LLM incorrectly imagines a scenario where the man is still walking to the same destination, but now he is walking over a bed, which makes it incorrectly

---

[6]The full dataset was released on Github on August 2021, so it is after the training cutoff for text-davinci-003 model. But it is a month before the cutoff for gpt-3.5-turbo. A smaller subset of inputs was released as a part of BIG-Bench in May 2021 with a canary string for LLMs to avoid including it in their training data.

conclude that the man "will get late" (see Suppl. B.1). These results indicate the importance of minimizing ambiguity when interacting with an LLM–wherever possible, adding small additional details (e.g., "walking on his bed at home") can increase the chances of a correct answer.

**Results on a novel counterfactual dataset.** In addition to the CRASS benchmark, we consider a counterfactual dataset that was created after the LLMs training cutoff date of September 2021. Li et al. (2023) develop a new synthetic dataset that was first uploaded on Github in 2022. We consider only the *counterfactual* variant of their dataset. It includes counterfactual statements of the kind: premise, hypothesis, conclusion. For each premise, there are two statements (one correct, one incorrect). We reformat the dataset to match CRASS format. That is, we create a multiple-choice question prompt as: premise; statement1, statement2. For example, here is an instance from the dataset: "If Charles Darwin had worked on curing people, A. Darwin would have written a paper on Aspirin; B. Darwin would have written a paper on species." Li et al. measure whether a model ranks the correct answer (A) higher than the incorrect answer (B).

We follow the exact same prompt text used for the CRASS dataset and evaluate GPT-4 (see Table 17 in Appendix A.3). GPT-4 obtains an accuracy of 88.6%, which is significantly higher than the best-reported accuracy from Li et al. (71.3% for GPT-3). This indicates that models like GPT-4 can generalize counterfactual arguments to new scenarios.

## 4.2 Inferring token causality: Necessary and sufficient causes

Given the promising performance of LLMs on a counterfactual inference task, we now investigate whether LLMs can infer whether events are necessary or sufficient for a focal event.

### 4.2.1 Motivating Example: Necessary and sufficient causes

To motivate, we present the following example story of a bottle of beer spilling at a party, a variation on the story presented in Henne & O'Neill (2022). Since the story was published in 2022, it cannot have been in the training set of either GPT 3 or GPT 4 models.

> PROMPT: Mike, Jack, Susan, and Peter are at a party. There is an open bottle of beer on the table. Mike is moving through the room and accidentally bumps against the table. Jack saw that the bottle was about to fall, and he reached out to catch the bottle. But just as Jack was about to catch the bottle, Peter bumped into Jack, which caused Jack to miss the bottle, which then fell to the ground. As a result, the bottle of beer fell to the floor, spilling the beer. Right after the bottle hit the floor, Susan tripped and fell onto the table.

Let us unwrap the story progressively to bring out the two key concepts of token causality: *necessity* and *sufficiency*.

**Necessary causality**: The property of an event, C, that if it had not occurred, then the outcome event, E, would not have occurred.

So we start by evaluating whether the LLM can make simple statements about necessary causality in the context of the beer bottle spilling story. First start by ask the LLM to provide a one sentence counterfactual explanation as to why Mike caused the spill.

> PROMPT: Mike, Jack, Ben, Susan, and Peter are at a party. There was an open bottle of beer on the table. Mike is moving through the room and accidentally bumps against the table. As a result, the bottle of beer fell to the floor, spilling the beer. Did Mike cause the beer to spill? Provide a one sentence counterfactual explanation as to why or why not.

Limiting the response to one sentence discourages more verbose responses.

> LLM: Yes, Mike caused the beer to spill; if Mike hadn't bumped into the table, the beer bottle would not have fallen and spilled.

The LLM zeroes in on the necessary causality as the explanation. More generally, we can prompt the LLM to invoke the necessary cause principle and output the token cause for an event.

In the above example, Mike's bumping into the table was necessary for the beer to spill. Moreover, it was also *sufficient* to cause the beer to spill. But there are many cases where a causal event is a necessary but insufficient cause of an outcome.

**Sufficient causality**: The property of an event, C, that if it had occurred, then the outcome event, E, would have occurred.

Formal models provide unambiguous definitions of sufficient causality; in Pearl's counterfactual notation, the probability a causal event C=1 is a sufficient cause is $P(E_{C=1} = 1|C = 0, E = 0)$ (Pearl, 2009b). In natural language this reads as "in cases where C was 0 and E was 0, what is the probability that E would have been 1 if C had been 1?"

But this phrasing awkward to express in situations where C and E were both indeed 1, not 0. So we turn to cognitive science literature to find a more natural text expression of sufficient cause. We use a variant of *robust sufficient causality* (Hitchcock, 2012; Woodward, 2006; Hitchcock, 2012):

**Robust sufficient causality**: Given that causal event C occurred, outcome event E would have occurred even if other events that contributed to B's occurrence had not occurred. To elicit a statement about robust sufficiency, we prompt the LLM to state what other causal events lead to the spill, and then predict a counterfactual about whether the spill would have occurred those events had not happened. This is a more challenging task than the necessary cause prediction, because it requires the LLM to solve a *causal frame problem* decide which other causal events to, some of which may their interactions.

PROMPT: Based on the information in the passage, what other events directly contributed to the fall of the beer and what would have happened if those events hadn't happened?

LLM: Other than Mike bumping into the table, the presence of the open beer bottle on the table directly contributed to the fall of the beer; if the bottle had not been placed on the table or had been properly closed, it would not have fallen and spilled even if Mike had bumped into the table.

This response highlights the capability of the LLM to set the *causal frame*, i.e., select causal events relevant to the inference problem. We did not specify which causal events were eligible for consideration. For example, the LLM could have included the presence of the table as a causal event, but it did not. This is in contrast to formal models such as SCMs where all the variables must be articulated in advance of running an inference query on the model. We conjecture that, as a probability model, the LLM is selecting probable causal abstractions based on both the prompt and statistical regularities in the training data. This allows its causal argument generation to adapt based on the scenario; if Mike bumping the table is the only causal event in consideration, it is sufficient, but that may change if other causal events are present. This highlights the flexibility of LLM's in causal analysis relative to SCMs and other formal models.

### 4.2.2 Evaluating LLMs on inferring necessary and sufficient causes

When the principle of necessity and sufficiency yields the same cause, token cause of an event can be easily determined. However, in many real-world scenarios, they may output different answers. For example, in the beer spilling example above, suppose there were two beer bottles on the table. Then the presence of any beer bottle on the table is a sufficient cause for beer spilling, but not necessary. Even if one of the bottles was not placed on the table, the other one would have spilled. Therefore, inferring necessary and sufficient causes typically requires applying the definitions of necessity and sufficiency and using formal reasoning (Halpern, 2016).

Below we investigate whether LLMs can generate causal arguments directly using the natural language description of an event. We evaluate on a collection of 15 vignettes (example scenarios) from Kueffner (2021). Table 10 shows sample vignettes. These vignettes are widely used to discuss and critique token causality definitions in the literature and span challenging scenarios across seven different types: symmetric overdetermination, switch, late preemption, early preemption, double preemption, bogus preemption, short circuit, and other miscellaneous examples. For example,

| Vignette Type | Input Context | Event | Actor | Nec. | Suff. |
|---|---|---|---|---|---|
| Overdetermination | Alice (AF) and Bob (BF) each fire a bullet at a window, simultaneously striking the window, shattering it (WS). | window shattering | Alice | No | Yes |
| Switch | Alice pushes Bob. Therefore, Bob is hit by a truck. Bob dies. Otherwise, Bob would have been hit by a bus, which would have killed him as well. | Bob's death | Alice | No | Yes |
| Late preemption | Alice (AF) and Bob (BF) each fire a bullet at a window. Alice's bullet hits the window first (AH). The window shatters (WS). Bob's bullet arrives second and does not hit the window (BH). | window shattering | Alice | No | Yes |
| Early preemption | Suppose Alice reaches out and catches a passing cricket ball. The next thing on the ball's trajectory was a solid brick wall that would have stopped the ball. Beyond that there was a window. | window being intact | Alice | No | Yes |
| Double preemption | Alice intends to fire a bullet at a window (AI). Bob intends to prevent Alice from hitting the window (BI). Bob tries to stop Alice (BSA). Bob is stopped by Carol (CSB). Alice fires a bullet (AF), hits the window (AH) and shatters it (WS). The window shatters (WS). | window shattering | Alice | Yes | No |
| Bogus preemption | Alice intents to put lethal poison into Carol's water. However, Alice does not put lethal poison into Carol's water (¬AP). Bob puts an antidote into Carol's water (BA). The water is lethal (L), if the poison is added without the addition of an antidote. If Carol would consumes the lethal water she would die (CD). Carol consumes her water (CC). Carol does not die (¬CD). | Carol's survival | Alice | No | Yes |
| Short circuit | Carol is alive (CA). Alice puts a harmless antidote in Carol's water (AA). Adding ant idote to the water, protects it against poison (WS - 'water save'). If Alice puts the antidote into Carol's water, B ob will poison the water (BP) Adding poison to an unprotected water makes it toxic (WT). If Carol would drink toxic water she would die (i.e. inhibiting CS). Carol consumes her water and survives (CS). | Carol's survival | Alice | No | Yes |
| Miscellaneous | If there is hot weather, flowers will die. Watering prevents the flowers to die in hot weather. The neighbor does not water the flowers in her yard. The flowers die. | flowers' death | neighbor's inaction | Yes | Yes |

Table 10: Example vignettes for evaluation of inferring necessary and sufficient causes, categorized by their *type* based on the different ways in which potential causes can interact to yield the final outcome. Each vignette tests two questions: *"Is {Actor} a necessary cause of {Event}?"* and *"Is {Actor} a sufficient cause of {Event}?"*.

symmetric overdetermination is a scenario where multiple processes (causes), all of which producing the same outcome (event), terminate at the same time. Suppl. C.1 provides a description for each type.

For each vignette, our goal is check whether an LLM can correctly identify whether a cause is necessary, sufficient, or both. To do so, we use the following prompt template, where PRINCIPLE is replaced by either "minimal change" (corresponding to necessary cause) or "multiple sufficient causes" (corresponding to sufficient causes). In each example, we provide two options: one of them is the necessary cause and the other the sufficient cause. Interestingly, the prompt itself was constructed using the gpt-3.5-turbo (see Suppl. C.2 for details). An example prompt for the early preemption vignette is shown below.

SYSTEM: You are an expert in counterfactual reasoning. Given an event, use the principle of [PRINCIPLE] to answer the following question.

PROMPT: Suppose Alice reaches out and catches a passing cricket ball. The next thing on the ball's trajectory was a solid brick wall that would have stopped the ball. Beyond that there was a window.

Is Alice a [NECESSARY/SUFFICIENT] cause of window being intact?

After your reasoning, provide the final answer within the tags <Answer>Yes/No</Answer>.

Using this prompt, we evaluate all 15 example scenarios on two LLMs: gpt-3.5-turbo and GPT-4. For example, on this vignette, both LLMs correctly answer that Alice is not a necessary cause; even without her action, the brick wall would have stopped the ball.For evaluating sufficiency, gpt-4 correctly answers that Alice is a sufficient cause of the window being intact, but gpt-3.5-turbo answers incorrectly (see responses in Suppl. C.4.1).

Tables 11 and 12 summarize the results over all vignettes. We see a significant difference between the accuracy of gpt3.5-turbo and gpt-4, indicating a marked change in ability between the two models. gpt-3.5-turbo fails to capture the nuances of necessity and sufficiency definitions and obtains an accuracy near random guess. However, gpt-4 is accurate

| Vignette Type | Necessary | Sufficient |
|---|---|---|
| ***gpt-3.5-turbo*** | | |
| Overdetermination | ✓, ✓ | X, ✓ |
| Switch | X,X | ✓,X |
| Late preemption | X | X |
| Early preemption | X, ✓, X | X, X, ✓ |
| Double preemption | ✓ | ✓ |
| Bogus preemption | ✓ | X, |
| Short circuit | X | X |
| Miscellaneous | X, ✓, ✓, X | ✓, ✓, X, ✓ |
| Total Accuracy | 46.6% | 46.6% |
| ***gpt-4*** | | |
| Overdetermination | ✓, ✓ | ✓, ✓ |
| Switch | ✓, ✓ | ✓, ✓ |
| Late preemption | ✓ | ✓ |
| Early preemption | ✓, ✓,✓ | ✓, |
| Double preemption | ✓ | X |
| Bogus preemption | ✓ | ✓ |
| Short circuit | X | X |
| Miscellaneous | ✓,X, ✓,✓ | ✓,✓, ✓,✓ |
| Total Accuracy | **86.6%** | **86.6%** |

Table 11: Accuracy of gpt-3.5-turbo and gpt-4 on inferring necessary or sufficient cause on 15 standard vignettes. The vignettes are divided into eight types (e.g., Early Preemption type has three vignettes). Each (✓/X) corresponds to a correct/incorrect answer on a single vignette. gpt-3.5-turbo fails at the task (worse than random chance) but gpt-4 can infer necessary and sufficient cause with high accuracy.

for most vignette types. This is remarkable because the LLM was not explicitly provided definitions of necessary and sufficient causes. As with the case of causal graph generation, however, the high accuracy also comes with unpredictable failure modes. For example, even as gpt-4 use the correct sufficiency principle for most vignettes, on the short circuit vignette (see Table 10), gpt-4 uses only counterfactual dependence principle to answer the question and fails to reason about the sufficieny of cause. The input prompt and incorrect response are shown in Suppl. C.4.2.

Since the above vignettes or their variants have been quoted in many publications, there is a concern that LLMs may have memorized them and may not be producing new inferences about the answers. We therefore report results on the "lab-vignettes" dataset, a new dataset we created that adapts the base vignettes to a laboratory setting. We do so by keeping the same type of scenario (e.g., short circuit), but changing the situation to that in a laboratory with reagents, test tubes, mixtures and crystals. Suppl. C.3 provides details on this dataset. On this new, unseen dataset, we find a similar pattern when comparing models. gpt-4 obtains significantly higher accuracy than gpt-3.5-turbo. While gpt-4 retains its overall accuracy, we find an interesting pattern: for both models, necessity is an easier concept to answer than sufficiency. gpt-4 obtains over 92% accuracy on deciding necessity of cause compared to 78% on deciding its sufficiency. This may be because necessity always involves comparing the output under a counterfactual world where only the cause is flipped; whereas sufficiency can be nuanced, involving counterfactual flips to all relevant other variables (and deciding what those variables should be).

Overall, our results indicate the capability of LLMs to understand a scenario in natural language and output token causes according to a pre-specified definition, but also show the lack of robustness due to unpredictable failures.

| Vignette Type | Necessary | Sufficient |
|---|---|---|
| *gpt-3.5-turbo* | | |
| Overdetermination | ✓, ✓ | X, ✓ |
| Switch | X,✓ | ✓,X |
| Late preemption | X | ✓ |
| Early preemption | ✓, X | X, X |
| Double preemption | ✓ | ✓ |
| Bogus preemption | ✓ | X |
| Short circuit | X | X |
| Miscellaneous | ✓, ✓, ✓, X | ✓, X, X, ✓ |
| Total Accuracy | 64.2% | 42.8% |
| *gpt-4* | | |
| Overdetermination | ✓, ✓ | ✓, ✓ |
| Switch | ✓, ✓ | X, ✓ |
| Late preemption | ✓ | ✓ |
| Early preemption | ✓, ✓ | X |
| Double preemption | ✓ | ✓ |
| Bogus preemption | ✓ | ✓ |
| Short circuit | ✓ | ✓ |
| Miscellaneous | ✓,X, ✓,✓ | ✓,✓, X,✓ |
| Total Accuracy | **92.8%** | **78.5%** |

Table 12: Testing dataset memorization issues with a novel "lab-vignettes" dataset. The average accuracy of gpt-4 stays the same as in the std vignettes, indicating that gpt-4's capabilities to infer necessary and sufficient cause can generalize to new data. Inferring necessary cause (93%) emerges as an easier task than inferring sufficient cause (78%).

### 4.2.3 Assessing responsibility

Using LLMs to produce causal arguments allows for a more nuanced understanding of causality, taking into account factors such as intention, moral obligation, and epistemic state that are difficult to capture using formal SCM approaches. We present a study of these capabilities in Supplementary D.

### 4.3 Evaluating LLMs ability to infer normality

Normality is a key element of evaluating token-causality. When evaluating the role an event had in causing some outcome, it is difficult to use a formal model to quantify the normality of that event. LLMs enable us to rely on common-sense knowledge to quantify how normal/abnormal an event was relative to an outcome. In this section, we describe an experiment designed to quantify the role normality plays in predicting the outcome to token causality problems.

For this analysis we used vignettes from the causal judgment benchmark in BIG-Bench Hard (Suzgun et al., 2022). The causal judgment benchmark collects examples from vignettes similar to those in Kueffner (2021), but are in more general language and provide more background information. Each of the vignettes ends with a yes/no question about whether a particular event caused the outcome. The benchmark provides ground truth labels for each question, collected from human annotators. The vignettes vary factors that drive human causal judgments, such as whether the causal event was necessary, sufficient, normal, whether it was causation by omission, whether it lead to a harmful outcome, etc. It is an ideal benchmark for evaluating how well LLMs would do in addressing practical token causality problems, as opposed to more contrived examples.

The benchmark has proven resilient to saturation as new generations of LLMs are released. For example, while several benchmarks in BIG-Bench Hard saw a dramatic increases in accuracy between GPT3 and GPT4, causal judgment had a small increase (61.5% in text-davinci-003 and has an accuracy of 67.6% in GPT-4-32K (AML-BABEL, accessed 2023-04-25). Top human performance is at 100% accuracy.

**Constructing a benchmark for normality.** We filtered out the vignettes that ask whether an agent intentionally caused an outcome and only focus on examples that ask if an event or agent caused an outcome. We do this because assessing intention is a separate task from ascribing causality, and because many intention examples were contrived (e.g., trolley problems) such that normality does not apply. We then manually reworded each vignette in order to mitigate against the possibility of the vignettes having appeared in the training. Each vignette was specifically designed to include sets of the factors that drive human causal judgments (necessity, normality, etc.). So we reword the vignette by applying a minimalist *lexical substitution* of words in the vignette so that we avoid changing those factors. To illustrate, the following is a vignette from the benchmark, which originally appeared in Knobe & Fraser (2008):

> The receptionist in the philosophy department keeps her desk stocked with pens. The administrative assistants are allowed to take the pens, but faculty members are supposed to buy their own. The administrative assistants typically do take the pens. Unfortunately, so do the faculty members. The receptionist has repeatedly emailed them reminders that only administrative assistants are allowed to take the pens. On Monday morning, one of the administrative assistants encounters Professor Smith walking past the receptionist's desk. Both take pens. Later that day, the receptionist needs to take an important message... but she has a problem. There are no pens left on her desk. Did Professor Smith cause the problem?

We reworded this vignette as follows, with the main changes in bold:

> The **executive assistant** in the **marketing department** keeps **notepads in his cabinet**. The **office assistants** are allowed to take the **notepads**, but **managers** are supposed to buy their own. The **office assistants** typically do take the **notepads**. Unfortunately, so do the **managers**. The **executive assistant** has repeatedly emailed them reminders that only **office assistants** are allowed to take the **notepads**. On **Tuesday** morning, one of the **office assistants** encounters **Manager Johnson** walking past the **executive assistant's cabinet**. Both take **notepads**. Later that day, the **executive assistant** needs to take an important **note**... but **he** has a problem. There are no **notepads** left **in his cabinet**. Did **Manager Johnson** cause the problem?

We then manually label each example as "normal" or "abnormal". In most cases these labels were not subjective assessments on our part. Each element in the causal judgment data set includes a citation to the source of the passage. In many of these studies, normality was explicitly varied across sections of a vignette, and thus we could confirm our labels from their source papers. However, in some cases, we had to make subjective best-guess assessments.

**Prompting the LLM.** We follow a two-step process. We first prompt the LLM to extract the causal event in question.

PROMPT: The last sentence of the following passages is a question. The question either ask if an outcome event happened because a certain causal event occurred. When the question asks specifically about "intention," the causal event is not the outcome, it is the action that lead to the outcome. State the causal event being asked about. Do not include the outcome event in the statement. Do not include other causal events in the passage.

Passage: The executive assistant in the marketing department keeps notepads in his cabinet. The office assistants are allowed to take the notepads, but managers are supposed to buy their own. The office assistants typically do take the notepads. Unfortunately, so do the managers. The executive assistant has repeatedly emailed them reminders that only office assistants are allowed to take the notepads. On Tuesday morning, one of the office assistants encounters Manager Johnson walking past the executive assistant's cabinet. Both take notepads. Later that day, the executive assistant needs to take an important note... but he has a problem. There are no notepads left in his cabinet. Did Manager Johnson cause the problem?

LLM:   Manager Johnson took notepad from the executive assistant's desk.

We then ask the LLM to evaluate the normality of the causal event in question. Inspired by both model-based and subjective notions of normality, we construct the following prompt.

PROMPT:   Given a causal event that causes an outcome event, the causal event is "abnormal" if any of the following is true:

- The causal event's occurrence was unexpected, unlikely, surprising, rare, or improbable
- The causal event was an agent's action or lack of action that non-accidentally, knowingly, or negligently violated social, legal, or ethical norms.

The opposite of "abnormal" is "normal". A causal event is 'normal" if its occurrence was expected, likely, probable, and not surprising. In cases where the causal event is an agent's action or lack of action, the action or lack of action can also be considered "normal" if it intentionally compliant with social, legal and ethical norms. The normality of a causal event is independent of whether the outcome event was bad or and whether the outcome event was normal.

Please read the following passage and statement about a causal event from the passage. Provide your step-by-step rationale as to why the causal event is abnormal or normal. Then, return the word "normal" if you think the causal event is normal or "abnormal" if you think the causal event is abnormal.

Passage: [PASSAGE]

Event: Manager Johnson took a notepad from the executive assistant's cabinet.

LLM:   Manager Johnson's action of taking a notepad from the executive assistant's cabinet is abnormal because it violates the established norm that managers are not allowed to take notepads from the executive assistant's cabinet. Despite the executive assistant sending reminders about this rule, Manager Johnson still took a notepad, making his action unexpected and unethical.

In defining normality, we attempted to capture both statistical norms (with keywords "expected," "likely," "probable," "frequent," "unsurprising," and their opposites) and prescriptive norms. Violation of prescriptive norms can depend on intention, so we use keywords "nonaccidentally", "knowingly," "negligently," and "intentionally." We also ask the model to provide a chain-of-thought "step-by-step" rationale for its answer.

**Results.**   Initial results show accuracy in inferring normality was 69.2% on gpt-3.5-turbo and 71.1% on gpt-4, which are comparable to performance on the core benchmark task. Table 13 shows three examples of correct and incorrect answers across answers by GPT-3.5-turbo and GPT-4.

Table 13 demonstrates the capability of LLMs to parse normality but also provides intuition for how the LLMs can get normality wrong. While results did improve between versions of gpt-3.5-turbo and gpt-4, there were some cases were gpt-4 was wrong when gpt-3.5-turbo was right, although its rationale when it was wrong was typically more persuasive.

## 4.4   Discussion

Formal approaches to token causality have struggled to represent background common sense elements of a causal judgment, such as necessity, sufficiency, normality and responsibility. We show through analysis of the CRASS (Frohberg & Binder, 2022) and causal judgment (Ghazal et al., 2017) benchmarks and related vignettes from Kueffner (2021) that LLMs perform well in generating causal arguments with these elements. Relatively lower performance for direct and chain-of-thought answering on the causal judgment benchmark highlight an opportunity to research and engineer systems that guide LLMs in assembling these core elements into solutions for practical problems in token causality questions.

| Passage and Question | Correct Answer | gpt-3.5-turbo rationale | GPT-4 rationale |
|---|---|---|---|
| Megan and Sarah work for the same organization. They each need to use a printer for tasks sometimes. Unfortunately, the printer isn't very reliable. If two people print at the same time, it usually jams. So the organization decided to institute an official rule. It declared that Megan would be the only one permitted to use the printer in the mornings and that Sarah would be the only one permitted to use the printer in the afternoons. As expected, Megan used the printer the next day at 9:00 am. But Sarah decided to ignore the official rule. She also used the printer at 9:00 am. The printer jammed immediately. Did Sarah cause the printer to jam? | Sarah using the printer at 9:00 am is an abnormal event because she followed policy. | Using the printer at 9:00 am is consistent with the official policy, making this causal event normal. **(incorrect)** | Sarah's decision to use the printer at 9:00 am despite the official policy is abnormal, as it intentionally violates the company's policy put in place to prevent the printer from jamming. This action is unexpected and goes against the established social and workplace rules. **(correct)** |
| The executive assistant in the marketing department keeps notepads in his cabinet. The office assistants are allowed to take the notepads, but managers are supposed to buy their own. The office assistants typically do take the notepads. Unfortunately, so do the managers. The executive assistant has repeatedly emailed them reminders that only office assistants are allowed to take the notepads. On Tuesday morning, one of the office assistants encounters Manager Johnson walking past the executive assistant's cabinet. Both take notepads. Later that day, the executive assistant needs to take an important note... but he has a problem. There are no notepads left in his cabinet. Did Manager Johnson cause the problem? | The office assistant taking the notepad is a normal event because they followed policy. | Taking notepads is expected behavior among both office assistants and managers according to the rules set by the executive assistant. Therefore, the causal event of both the office assistant and Manager Johnson taking notepads is normal. **(incorrect - for managers, this is a norm violation)** | Taking a notepad from the executive assistant's cabinet is abnormal for the office assistant in this situation because they contributed to the problem of the executive assistant having no notepads left, even though it is typically allowed for them to take notepads. However, it is important to note that the managers action of taking a notepad is even more abnormal and against the established rules. **(incorrect - the normality of the office assistant taking the pen is independent of the outcome)** |
| Mark was about to go pick up his daughter from a daycare in the city. In the parking lot, he stopped to chat with friends. Because of this delay in picking up his daughter, Mark's friend, who waited 15 min for Mark to arrive, took the child home in his car. On the way home, the friend's car was hit by a reckless driver. Mark's daughter, the only seriously injured victim, suffered severe arm injuries. Did Mark cause his daughter's injury? | While stopping to talk to friends is normal, doing so to the point of being late and causing others to wait is a social norm violation. The cause of Mark's being late is therefore abnormal. | Stopping to talk with friends in a parking lot is a common and expected behavior, making this causal event quite normal. **(incorrect)** | Marking stopping to talk with friends in the parking lot is a normal event as people often engage in casual conversations when they encounter acquaintances. It is not unexpected, unlikely, or a violation of social norms. **(incorrect)** |

Table 13: Comparative assessments of normality between gpt-3.5-turbo and gpt-4. Stories taken from the BIG-Bench causal judgments task.

# 5   A New Frontier for Causality

Our results on evaluating large language models for causal DAG generation, counterfactual reasoning, and token causality demonstrates that they bring significant new capabilities across a wide range of causal tasks. We believe that these capabilities signal the beginnings of a new frontier and will transform the practice and research of causality. We start by summarizing key takeaways on how to interpret the new capabilities that LLM bring to causality. We then provide implications for practitioners and future research directions at the intersection of LLMs and causality.

**What is new with causality and LLMs?**

- LLMs provide access to domain knowledge that was, heretofore, only available via human domain experts. LLMs can provide this domain knowledge when explicitly asked, and also implicitly, e.g., as we see in background knowledge necessary for token causality vignettes.

- LLMs provide a flexible, natural language-based interaction mechanism for conducting causal analysis that can work alongside existing tools, thus democratizing access to causal tools and knowledge.

- LLMs offer a new capability to extract the key primitives of an token causality question (necessity, sufficiency, normality, etc.). This opens up the possibility of building systems based on formal token causality research for practical scenarios for perhaps the first time.

It is important to recognize that answering causal questions require iterative steps in practice. LLMs can potentially enable a wide and effective adoption of causality in practice and break the barriers between different research communities related to causality.

**What is not changing with causality and LLMs?**

- Especially for high-risk and high-value tasks, people have long relied on rigorous analyses in various formal frameworks to ensure correctness of their decision-making. The need for rigorous, well-documented, and verifiable analyses has not changed. This implies that LLM-driven applications must have tight integration with more formal approaches to causal reasoning. LLMs should be able to fluidly connect their language generation with the use of formal tools, including re-framing or translating natural language scenarios to formal descriptions, and ensuring their natural language responses are in accordance with the conclusions of formal analyses.

## 5.1 Implications for Causality Practitioners

**Augmenting human expertise with LLMs.** LLMs can enable effective human-AI interaction that reduces human burden during causal analyses, while ensuring trustworthiness and human verifiability. One of the key reasons causal methods are not widely deployed is because using them requires expertise in expressing the causal assumptions formally through a graph and then verifying them through robustness checks. LLMs can act as an assistant in both these processes, augmenting the capability of a practitioner and helping reduce friction in starting a causal analysis.

Given the challenging task of algorithmic causal discovery, most practitioners start their causal analysis by constructing a graph manually based on their domain knowledge. Rather than creating a graph from scratch, we recommend that practitioners input their dataset metadata to an LLM and obtain an initial version of the graph that they can iterate on. At least for medium sized graphs, our results indicate that LLMs can match state-of-the-art accuracy in learning real-world graphs, as long as there is descriptive metadata available for each variable. Doing so can save time since editing a graph can be faster than coming up with all edges. Alternatively, given that LLMs obtain their best accuracy in pairwise causal discovery tasks, another way is use LLMs to critique human-generated graphs. LLMs can be used to iteratively critique the presence or absence of edges and help the analyst discover any missing edges.

Apart from graph creation, the other challenging task is to validate the output of any causal analysis, since the output depends on the initial causal assumptions (e.g., the conditional independence assumptions implied by the graph). Here LLMs can act as useful assistants and help the analyst through chat conversation towards finding any flaws in the research and/or planning a robustness test. LLMs can also help in suggesting specific robustness checks for a causal analysis, such as the variables to use as negative controls. In Suppl. F.2, we present a case study of a conversation with an LLM to identify negative controls for an observational study of vaccine efficacy.

**Applying LLMs to novel scenarios.** We evaluated LLMs on a combination of known, popular datasets (that are likely to be in LLMs' training data) and datasets developed after LLM's training cutoff date (that are not present in the LLMs' training data). On both kinds of datasets, we obtain similarly high accuracies across different causal tasks, indicating that the presence of the specific dataset in the LLMs' training data is not a prerequisite for doing well on a causal task. In our experiments with datasets outside the LLMs training data, we expect that the LLM's capabilities are a result of transforming relevant domain knowledge extracted from other, older texts in the training data.

We do not expect an LLM to construct causal assumptions or causal graphs on new concepts beyond what is included in its training corpus. That said, even analyses of novel causal relationships generally occur in the context of other, better understood causal mechanisms. Therefore we expect practitioners to benefit from using LLMs for inferring at least a subset of cause-effect relationships in real-world datasets. For scenarios entirely outside an LLM's training corpus, the use of retrieval augmented generation presents an opportunity for future research.

The correctness of causal assumptions is critical for the validity of causal analyses, e.g., of numerical data for scientific discovery or other novel scenarios. Until now, the sole advice for ensuring the correctness of causal assumptions was to advise practitioners to take care and consult with domain experts. The construction of causal assumptions or causal graphs and the criticism of practitioner's assumptions based on LLM's output presents a significant advance in technological support for this critical task.

**LLM + Causal tools: A new pipeline for causal analysis**   While LLMs can help with framing and validating causal assumptions—the steps in a causal analysis where human expertise is central—they are not capable of replacing the tools that do statistical analysis for estimating causal quantities. These tools (e.g., DoWhy, EconML, Ananke) implement principled algorithms from graphical causal models and the potential outcomes literature and are a workhorse for a causal analyst. We envision that LLMs can enable an end-to-end pipeline for causal inference where the code control seamlessly transfers between an LLM and causal tools. Practitioners can construct pipelines where graph creation is done in LLM-assisted interfaces while the final output graph is passed in to a statistical inference tool. Given a graph, an LLM may also be used to generate the code for conducting downstream analysis with a specified causal tool. Based on recent research (Schick et al., 2023), it may also be possible to augment LLMs and enable them to call out relevant causal tools and return the answer in a conversational chat interface. Suppl. F.1 presents a case study conversation with GPT-4 asking it to create sample code for synthetic data generation and causal analysis using the PyWhy open-source libraries.

**LLMs as a fluid conversational interface, merging covariance- and logic-based reasoning.**   In practice when tackling complex causal challenges, people strategically alternate between different modes and methods for causal reasoning. People may begin with a logical reasoning approach to frame a high-level problem, then toggle to a covariance based approach to solve a subtask. They may switch to a mental simulation to consider counterfactuals, or validate consequences against logical assumptions. And they may continue interchanging complementary analytical methods repeatedly before honing in on their final answer. This kind of fluid iteration between modes of thinking to date required humans to translate their premises and knowledge across frameworks. Now, LLMs may be able to do more of that.

If so, this may allow a merging of the many kinds of causality into a single higher-level abstraction, allowing people to ask and have answered a much broader and higher-level set of causal questions than could have been addressed through computational methods previously. A first step towards this goal can be to include the information about variables into the problem of generating causal DAGs.

## 5.2   Ethical Implications and Societal Impacts

Causal modeling is integral to AI interpretability and fairness. It provides explanations behind an AI's predictions in individual cases, enhancing interpretability and fairness in AI decision-making processes. However, without stringent understanding and control over how LLMs formulate causal arguments, there's a risk of these models being manipulated to produce seemingly valid yet misleading explanations for its explanations. Such misrepresentations could falsely attest to the fairness of a decision or rationalize its basis, potentially leading to misplaced trust and unjust outcomes under the guise of algorithmic neutrality and causal rigor.

The field of causal inference has focused on theoretical and empirical analysis of real-world causal mechanisms, including in high-stakes areas like policy-making and healthcare. Despite its numerical basis, this field is not immune to subjective influences. For instance, a causal analysis of how 'race' affects 'wealth' relying on rigorous causal identification and high quality data would still depend on a definition of race – a nebulous concept impossible to randomize in an experiment and which varies across cultures and history. One of the strengths of traditional causal inference analysis is that the subjective elements of the analysis are explicit in the specification of the model, such as the choice of variable definition and the edges in a DAG. When these assumptions and biases are explicit, they can be scrutinized. But with LLMs, these assumptions and biases, shaped by diverse and sometimes highly problematic internet data, are hidden in the black box of learned weights in a transformer architecture. Even these weights are often concealed behind proprietary boundaries. This lack of transparency could lead to the misconception that LLMs' causal

claims are unbiased due to their algorithmic nature. This is concerning given claims that LLMs learn *world models* (i.e., causal models). Indeed, an LLM can be fine-tuned to reflect reflect *any* viewpoint (e.g., vaccine skepticism). It is imperative that in researching the causal capabilities of LLMs, we avoid creating a veneer of causal infallibility.

## 5.3 New research questions on causality and LLMs

Our work is a proof of concept that LLMs hold the potential to provide domain knowledge and enable natural and flexible interactions for causal tasks. It will require further advances to enable effective human-AI interaction to realize this potential.

**Knowledge-based causal DAG generation.**   We acknowledge that our formulation of knowledge-based pairwise edge inference is different from the standard formulation. However, given that LLMs demonstrate non-trivial performance of inferring causal relationships based on variable names alone, we believe that this can have substantial implications on research on algorithmic causal discovery and, more generally, causal model specification. Specifically, these results beg the question of how we can reformulate the problem of specifying a causal DAG that best leverages large language models and existing knowledge in large amounts of texts.

Graphical causal models require the modeler to specify a causal graph. They do this using domain knowledge about which variables in the domain cause witch – i.e., the modeler leverages *meta-information* about the variables to construct the graph. In contrast, the canonical problem formulation of causal discovery aims to address this task with algorithms that attempt to reverse engineer the causal graph using covariance in observational or experimental data. The attractiveness of this approach is that relying on statistical evidence in the data helps avoid subjective model misspecification errors. However, this approach completely ignores meta-information about variables. Our results indicate that this may be a mistake since many datasets provide both data values and meta-information that can be useful for inferring at least a part of the full graph. One promising direction is to consider the output of an LLM as a prior for a causal discovery algorithm. The relevant question is to what extent having an LLM-based prior would increase the accuracy of causal discovery algorithms, where (Choi et al., 2022) present some promising results. Another, more provocative direction is to recognize that current discovery algorithms struggle on real-world datasets and we can aim for redefining the causal discovery problem such that it includes the meta-information about variables and the existing knowledge encoded through LLMs (Zhiheng et al., 2022). Developing new discovery algorithms that make use of both meta-information and covariance in the data can be a fruitful research direction, wherein the LLM may be used as a prior, as a critic during learning, and/or as a post-processor.

In addition to building new algorithms, LLMs can also facilitate building benchmarks for causal discovery algorithms. As illustrated from our results, causal discovery benchmarks are usually created by close collaboration with domain experts, with a strong assumption that domain experts know the groundtruth causal graphs. However, this assumption does not hold because 1) Possible edges grow superlinearly with the number of variables, so it is unlikely for a human expert to go over all possible edges reliably; 2) More importantly, domain experts may not be able to precisely provide the causal link between two variables given their knowledge. LLMs can help identify potentially missing edge and correct mislabeled edge, or at least point out relations that require additional checking.

**LLM-guided effect inference.**   Identifying the correct adjustment set (e.g., using the backdoor criterion) is one of the most important problems for causal effect inference. However, the validity of the backdoor criterion depends on access to the true graph. LLMs' capability for generating causal DAG can be useful for effect inference and opens up a research question on how to use LLMs to infer valid backdoor sets. In addition to discovering the full graph, techniques that utilize partial graph structure (Entner et al., 2013; Cheng et al., 2022) to identify backdoor sets may also benefit from the ability of knowledge-based DAG generation with LLMs to infer specific edges. Similarly, it may be interesting to see if LLMs can be used to suggest potential instrumental variables for a causal task, given metadata about observed variables.

A second promising direction is in utilizing LLM's domain knowledge to help build robustness and validation checks for a given causal analysis. Typically, this involves an analyst selecting appropriate robustness checks given a problem and then reasoning over the background context to identify the correct variables to use for the check. Can LLMs be

used for determining which robustness tests may be applicable for a given problem and which specific variables are most suited to be used in those tests?

**Systematizing token causality and attribution.** The ability to support token causal inference is one of the most potentially disruptive capabilities of LLMs. In domains ranging from law to intelligence analysis, analysts have to first summarize key elements of text data from multiple sources, then synthesize explanations about the degree to which events contributed to other events, i.e., explanations of why and how things happened. While it is well known LLMs excel at the former task, using LLMs for token causality presents a major opportunity to impact the latter. Using LLMs for token causality also has potential impact to root cause analysis and credit assignment in domains ranging from engineering to reinforcement learning.

Formal causal models for inferring token causality questions have struggled formalizing the many elements of common sense background knowledge that rely on when judging token causality (Halpern, 2016; Icard et al., 2017; Knobe, 2003; Henne et al., 2017). We've argued that LLMs can work with these background concepts directly in natural language, sidestepping the challenge of shoehorning them into a formal model.

The BIG-Bench causal judgment dataset (Suzgun et al., 2022) is ideal for testing LLMs ability to answer practical token causality questions. LLMs have performed less well on this benchmark relative to best human performance as compared to the rapid saturation of other benchmarks. However, answering these types of questions involves reasoning over commonsense background knowledge concepts, such as necessity, sufficiency, and normality. Our analysis shows that LLMs do better at answering questions about these basic elements than answering high-level token causal judgment questions directly. A promising direction of research is developing methods that guide LLMs to use these token causality primitives to answer higher level token causal judgment questions, perhaps using formal token causality theory as a guide.

**Human-LLM collaboration.** A recurring theme throughout the paper is the promise of human-AI collaboration in causal problems. There are interesting research questions on how best to facilitate such human-LLM collaboration for building graphs. The interaction will likely be iterative for the maximum impact: LLMs may suggest graph edges, take feedback from a human, and also can give feedback on a manually generated graph. Given that building a graph is a common impediment to doing causal analyses, an improvement in this task will have important consequences for the widespread adoption of causal data analysis.

**Understanding and improving how LLMs generate causal arguments.** Why LLMs demonstrate such causal capabilities, as well as the limits of these capabilities is not yet well understood. The LLMs certainly do well in interpolating between causal language in its training data to answer interventional and causal language. The question is whether they are limited to such interpolation Willig et al. (2023) or if they can learn a "world model" (i.e., a catholic causal model) Hao et al. (2023). According to the causal hierarchy theorem (Bareinboim et al., 2022), an LLM can only make interventional and counterfactual inferences outside of its training data with specific interventional and counterfactual inductive biases. Questions of whether an LLM entails such causal inductive biases, what exactly they might be, and for what types of queries can they be relied upon will require further study. At the same time, we see unpredictable failure modes, even in tasks where LLMs obtain high accuracy. In summary, understanding the nature of causal reasoning in LLMs and how to improve their robustness is a key future question.

## 6 Conclusion

Human domain knowledge has always been a central piece in causal analysis. In this paper, we studied the capabilities of large language models and found that LLMs can provide value by mimicking that domain knowledge, as a result of being trained on vast amounts of human-generated text. This mimicking is a result of a complex training process so it is not explainable and neither predictable: an LLM can fail on some queries while successfully building a causal arguments in others. What is remarkable is how few times that such errors happen: our evaluation finds that on average, LLMs can outperform state-of-the-art causal algorithms in graph discovery and counterfactual inference, and can systematize nebulous concepts like necessity and sufficiency of cause by operating solely on natural language input.

From a research perspective, these results open up more questions than they answer, and we provide a list of research questions at the intersection of LLMs and causality. At the same time, due to the demonstrated capabilities of LLMs, we foresee a marked impact of LLMs on the practice of causal analysis. We outlined how LLMs can help reduce the burden of human expertise in tasks like building the causal graph, effect inference, and attribution. Another contribution of LLMs is in bridging the divide between covariance- and logic-based causal analysis. By providing a flexible natural language interface for answering causal queries, LLMs can serve to unify these two branches of causal analysis and allow analyses that seamlessly straddle both to answer real-world causal questions.

## Acknowledgments

We thank all the people who gave us feedback on earlier versions of this work, as well as the thought-provoking questions and comments during seminars that helped refine our ideas. We also thank the anonymous reviewers whose comments helped improve the paper's contributions.

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

# A Causal graph generation: Additional details

**Model version.** For all experiments, we use gpt-3.5-turbo-0613 and gpt-4-0613 model versions.

## A.1 Example prompts for the Tubingen benchmark

| Two prompts per pair |
| --- |
| **Template**: 
 - Does changing {A} cause a change in {B}? Please answer in a single word: yes or no. 
 - Does changing {B} cause a change in {A}? Please answer in a single word: yes or no. 
 **Examples:** 
 - Does changing the altitude cause a change in temperature? Please answer in a single word: yes or no. 
 - Does changing the temperature cause a change in altitude? Please answer in a single word: yes or no. 

 - Does changing a ball's speed at the beginning of a ball track cause a change in its speed at the end of the track? Please answer in a single word: yes or no. 
 - Does changing a ball's speed at the end of a ball track cause a change in its speed at the beginning of the track? Please answer in a single word: yes or no. |
| **Single prompt** |
| **Template:** 
 - Which cause-and-effect relationship is more likely? 
 A. changing {A} causes a change in {B}. 
 B. changing {B} causes a change in {A}. 

 Let's work this out in a step by step way to be sure that we have the right answer. Then provide your final answer within the tags <Answer>A/B</Answer>. 
 **Example:** 
 - Which cause-and-effect relationship is more likely? 
 A. changing the altitude causes a change in temperature. 
 B. changing the temperature causes a change in altitude. 

 Let's work this out in a step by step way to be sure that we have the right answer. Then provide your final answer within the tags <Answer>A/B</Answer>. |

Table 14: Two kinds of prompt templates for the Tübingen benchmark. The first asks two questions per variable pair whereas the second ("Single prompt") asks a single question to orient each pairwise edge.

## A.2 Example LLM outputs for the Neuropathic pain dataset

An example of a correct output from gpt-3.5-turbo.

LLM: First, we need to understand the terms involved in the cause-and-effect relationship.

Left T6 Radiculopathy refers to a condition where there is damage or irritation to the nerve root that exits the spinal cord at the level of T6 on the left side of the body.

DLS T5-T6 refers to degenerative changes in the discs, ligaments, and/or bones at the level of T5-T6 in the spine.

Now, let's consider the two options:

A. Left T6 Radiculopathy causes DLS T5-T6. This would mean that the damage or irritation to the nerve root at T6 on the left side of the body somehow leads to degenerative changes in the discs, ligaments, and/or bones at the level of T5-T6. While it is possible that nerve damage could contribute to degenerative changes in the spine, it is more likely that the opposite is true.

B. DLS T5-T6 causes Left T6 Radiculopathy. This option suggests that the degenerative changes in the discs, ligaments, and/or bones at the level of T5-T6 are causing the Left T6 Radiculopathy. This is a more likely cause-and-effect relationship, as degenerative changes in the spine can lead to nerve root irritation or compression.

Therefore, the more likely cause-and-effect relationship is: <Answer>B</Answer>DLS T5-T6 causes Left T6 Radiculopathy.

An example of an incorrect output.

LLM: First, we need to understand what DLS T5-T6 and Right T6 Radiculopathy are. DLS stands for degenerative lumbar spondylosis, which is a condition where the spinal discs in the lumbar region of the spine degenerate over time. T5-T6 refers to the thoracic vertebrae in the middle of the spine. Radiculopathy is a condition where there is compression or irritation of a nerve root, which can cause pain, numbness, or weakness in the area of the body that the nerve supplies.

Based on this information, it is more likely that option B is the correct cause-and-effect relationship. This is because radiculopathy is typically caused by compression or irritation of a nerve root, which can then lead to pain or other sym ptoms in the area of the body that the nerve supplies. In this case, the Right T6 Radiculopathy is causing symptoms in the area of the body that corresponds to the DLS T5-T6 region of the spine.

Therefore, the correct answer is <Answer>B</Answer>.

### A.3 Novel datasets created after LLMs training cutoff date

**Novel Tubingen dataset.** Table 15 shows the novel pairs constructed from recently published books.

**Novel Alzheimer's dataset.** Table 16 shows graph discovery accuracy for gpt-3.5-turbo and gpt-4 compared to data-based discovery algorithms such as DAG-GNN and NOTEARS.

**Novel Counterfactual dataset.** Table 17 shows results of using LLMs like GPT-4 on the novel counterfactual dataset from Li et al. (2023). We compare GPT-4 to accuracy obtained by previous models evaluated by Li et al. (2023).

## B  Evaluation: LLMs and counterfactual reasoning

### B.1  Details on CRASS counterfactual evaluation

For computing accuracy, the LLM obtains a score of 1 if it outputs the correct answer option and 0 if it outputs an incorrect option. If the LLM output does contain an answer among the options A/B/C, we consider as if the LLM has skipped the question and assign a score according to the probability of obtaining the correct answer under random chance (0.33).

### B.1.1  Example prompt

Here is an example prompt.

| Category | Var1 | Var2 | Causal Direction | Book Title | Year |
|---|---|---|---|---|---|
| Biology | high temperature and high shear rate | protein denaturation | Right | Sustainable Protein Sources: Advances for a Healthier Tomorrow | 2023 |
| Biology | amount of algae in coral | color of coral | Right | Amaze your brains with 900+ Cool science Facts | 2023 |
| Finance | credit card score | on-time payments | Left | Bounce Back: The Ultimate Guide to Financial Resilience | 2023 |
| Medicine | Kidney damage | Cirrhosis | Left | Introduction to Global Health | 2022 |
| Science | grain size reduction | metal strength | Right | Materials Chemistry | 2023 |
| Civil Engineering | building flexibility | bearing displacement | right | Advances in Structural Dynamics | 2023 |
| Medicine | inflammatory bowel diseases | Mesenteric venous thrombosis | Right | Understanding Crime Through Forensic Sciences | 2023 |
| Health | Oral contraceptive pills | weight gain | Right | Balanced Life: Your Ultimate Weight Management Guide | 2023 |
| Soil Science | Usage of Nitrogen fertilizer | Soil pH | Right | Soils for nutrition: state of the art | 2022 |
| Technology | battery capacity | ambient temperature | Left | Battery Management System and Its Applications | 2022 |

Table 15: A sample of 10 pairs from the Novel Tubingen dataset.

| Algorithm | NHD | No. of Predicted Edges | Baseline NHD | Ratio |
|---|---|---|---|---|
| NOTEARS | 0.22 | 10 | 0.32 | 0.69 |
| DAG-GNN | 0.37 | 20 | 0.44 | 0.83 |
| gpt-3.5-turbo | 0.21 | 21 | 0.38 | 0.55 |
| gpt-4 | **0.14** | 25 | 0.48 | **0.28** |

Table 16: Results for the novel Alzheimer's dataset from Abdulaal et al. (2023). For data-based algorithms, we report the results for the best configuration. For LLMs, we report results with temperature=0.

| Algorithm | Accuracy |
|---|---|
| BERT | 34.2 |
| RoBERTa | 61.4 |
| MPNET | 66.9 |
| GPT-2 | 53.7 |
| GPT-3 | 71.3 |
| gpt-4 | **88.6** |

Table 17: Accuracy results for the novel counterfactual dataset. Results for models other than gpt-4 are from Table 2 in Li et al. (2023).

PROMPT: A woman sees a fire. What would have happened if the woman had touched the fire?
    A: She would have seen fire.
    B: That is not possible.
    C: She would not have been burned.
    D: She would have been burned.

    Provide your answer within the tags, <Answer>A/B/C/D</Answer>.

### B.1.2 Example responses from gpt-4

Here we list three example responses from gpt-4 that do not match the benchmark answer. In the first two examples, the answer from gpt-4 can be considered correct.

SYSTEM: You are a helpful assistant for counterfactual reasoning.

USER: A man catches a ball. What would have happened if the man had caught a water balloon?
    A: He would have caught the ball and not the balloon.
    B: He would have got wet.
    C: He would have died.
    D: He would have remained dry.

    Let's work this out in a step by step way to be sure that we have the right answer. Then provide your final answer within the tags, <Answer>A/B/C/D</Answer>.

GPT-4: Step 1: Identify the difference between the two scenarios.
In the first scenario, the man catches a ball. In the second scenario, the man catches a water balloon.

Step 2: Determine the consequences of the difference.
Catching a ball typically does not result in getting wet or any other significant consequences. However, catching a water balloon may cause it to burst, potentially getting the man wet.

Step 3: Evaluate the possible outcomes.
A: He would have caught the ball and not the balloon. (This outcome does not address the counterfactual scenario where the man catches a water balloon.)
B: He would have got wet. (This outcome is possible if the water balloon bursts upon being caught.)
C: He would have died. (This outcome is highly unlikely and not a reasonable consequence of catching a water balloon.)
D: He would have remained dry. (This outcome is possible if the man catches the water balloon without it bursting.)

Step 4: Choose the most likely outcome.
The most likely outcome is that the man would have either gotten wet or remained dry, depending on whether the water balloon burst or not. Since we cannot determine with certainty whether the water balloon would burst, we cannot definitively choose between options B and D.

<Answer>Uncertain between B and D</Answer>

SYSTEM: You are a helpful assistant for counterfactual reasoning.

USER: A man does not get very nervous. What would have happened if he had gotten very nervous?
    A: That is not possible.
    B: The man would have passed out.
    C: The man would not have passed out.

Let's work this out in a step by step way to be sure that we have the right answer. Then provide your final answer within the tags, <Answer>A/B/C</Answer>.

GPT-4:  Step 1: Identify the counterfactual situation - The man gets very nervous.
Step 2: Consider the possible outcomes of the counterfactual situation.
a) The man passes out due to extreme nervousness.
b) The man does not pass out but experiences other symptoms of nervousness.
Step 3: Evaluate the likelihood of each outcome.
a) Passing out due to extreme nervousness is possible, but not guaranteed.
b) It is more likely that the man would experience other symptoms of nervousness without necessarily passing out.

Final Answer: <Answer>C</Answer>

Below we show an example where people may easily understand that the man is walking on his bed indoors, but the LLM incorrectly assumes that he continues walking *outdoors* on a bed towards the same destination.

SYSTEM:  You are a helpful assistant for counterfactual reasoning.

USER:  A man walks on a street. What would have happened if a man had walked on a bed?
A: He would have been late.
B: Nothing special would have happened.
C: He would have arrived on time.

Let's work this out in a step by step way to be sure that we have the right answer. Then provide your final answer within the tags, <Answer>A/B/C</Answer>.

GPT-4:  Step 1: Consider the difference between walking on a street and walking on a bed.
- Walking on a street is a normal activity and allows for efficient movement from one place to an other.
- Walking on a bed is unusual and would likely slow down the person's movement due to the soft and unstable surface.

Step 2: Evaluate each option based on this difference.
A: He would have been late.
- This option suggests that walking on a bed would have slowed the man down, making him late for his destination.

B: Nothing special would have happened.
- This option implies that walking on a bed would not have had any significant impact on the man's journey.

C: He would have arrived on time.
- This option suggests that walking on a bed would not have affected the man's arrival time at his destination.

Step 3: Determine the most likely outcome.
- Given that walking on a bed is an unusual and inefficient way to move, it is most likely that the man would have been slowed down and potentially arrived late at his destination.

<Answer>A</Answer>

## C   Evaluation: Necessary and Sufficient Causes

### C.1   Classes of token causality vignette scenarios

For a discussion of the different scenario classes, see Chapter 3 in Kueffner (2021). Below we provide a brief summary.

1. Symmetric Overdetermination: Multiple processes that lead to the same outcome, complete at the same time.

2. Switch: An action chooses between multiple processes, each of which lead to the same outcome. Thus, the action becomes immaterial for the outcome.

3. Late preemption: Two processes are running in parallel. Both would produce the same outcome, but one completes before the other.

4. Early preemption: Two processes would produce the same outcome, but one completes before the other has started.

5. Double preemption: A focal process leads to an event. An action that would have stopped the focal process, is itself stopped by another process.

6. Bogus preemption: An action stops a focal process, but the focal process itself is inactive wrt its effect on the outcome.

7. Short circuit: An original action makes the focal process inactive. An action is taken to prevent this inactive process; it triggers the process but the process is unable to complete because of the original action.

## C.2 Prompt generation for necessary and sufficient causes

Given the complicated definitions of token causality (see, for example, definitions of "actual causality" in Chapter 2 of Halpern (2016)), it is non-trivial to express them in natural language. We thus turned to the LLM itself to suggest prompts to use.

Below we describe a method to generate an appropriate prompt for a given task, based on Zhou et al. (2022). For generating a prompt that instructs an LLM to identify the necessary cause, we provide a meta-prompt that gives a few demonstrations of the task (vignettes as input and the necessary cause as output) and then asks the LLM to generate the instruction for that task. We then use a paraphrased version of the LLM's instruction to create our prompt.

Specifically, we provide three example vignettes with the scenario and correct token cause and ask the LLM to output the instructions that must have led to the output in each example. After a few executions, the LLM provided a reasonable principle called "minimal rewriting". We tweaked the wording to the more common "minimal change" principle and use it in our prompt.

Here is the meta-prompt and its response from gpt-3.5-turbo.

META-PROMPT: I taught my friend some concepts in counterfactual reasoning. Based on my teaching, they provided the following output, given an input scenario and candidate outputs.

Input: A window shatters when it is hit by a bullet. Alice and Bob each fire a bullet at a window, simultaneously striking the window, shattering it.
Candidate outputs: Both Alice and Bob, Any of Alice or Bob
Output: Both Alice and Bob

Input: Alice, Bob, Carol, Dave and Eve all fire at a window. The window shatters after three hits.
Candidate outputs: All five individuals, any individual, the first three individuals who fired
Output: The first three individuals who fired

Input: Alice fills a sink with water. At each time interval Alice adds another drop of water. At one point the sink overflows.
Candidate outputs: All droplets added to sink, Only the last droplet added to sink
Output: Only the last droplet added to sink

What logic was the friend using to select the correct output?

GPT-3.5-TURBO: : *It seems that your friend was applying the principle of minimal rewriting to select the correct output. The principle of minimal rewriting is a criterion used in counterfactual reasoning to evaluate which changes to the initial situation are necessary to obtain a given outcome.*

| Vignette Type | Input Context | Event | Actor | Nec. | Suff. |
|---|---|---|---|---|---|
| Overdetermination | There is a fire in the chemistry lab. A can of water would douse the fire. Agents X and Y both spray a can of water each, dousing the fire. | fire being doused | Agent X | No | Yes |
| Switch | Reagent X is added to a mixture, which leads to an explosion and kills Sam. Otherwise, Reagent Y in Sam's pocket would have infected him and killed him as well. | Sam's death | Reagent X | No | Yes |
| Late preemption | Any of Reagent X or Reagent Y can be added to a mixture to convert it into a crystal. Reagent X is added first and the mixture turns to crystal. Reagent Y is added later and but does not mix since the crystal is already formed. | crystal formation | Reagent X | No | Yes |
| Early preemption | Sam reaches out and catches the falling test tube containing a contaminating reagent. The next thing on the test tube's trajectory was a soft foam mattress which would have prevented the test tube from reaching the floor. Beyond that there was the floor. | floor being contaminated | Sam | No | Yes |
| Double preemption | Sam intends to add reagent X to a mixture that would convert it to a crystal. Riya intends to prevent Sam from adding the reagent. Riya tries to stop Sam from adding the reagent, but Riya is stopped by Frank. Sam adds the reagent and the crystal is formed. | crystal formation | Sam | Yes | No |
| Bogus preemption | Sam intents to put Reagent X in a mixture to make it explode. However, Sam does not put the reagent in the mixture. Bob puts an anti-explosion Reagent Y into the mixture. The mixture would explode if Reagent X is added without the addition of Reagent Y. Mixture does not explode. | mixture not exploding | Sam | No | Yes |
| Short circuit | A test tube contains a mixture. Sam puts sand in the mixture. Adding sand to the mixture protects it against explosion. Only if Sam puts sand in the mixture, Riya will add Reagent Y to the mixture. Adding Reagent Y to a mixture makes it explode, unless the mixture already contains sand. There is no explosion. | avoiding explosion | Sam | No | Yes |
| Miscellaneous | Sam is heating Reagent X in a test tube. If the heat is not stopped at the right time, the test tube would explode. Sam does not stop the heat. The test tube explodes. | explosion | Sam's inaction | Yes | Yes |

Table 18: "lab-vignettes": Examples of novel vignettes for evaluation of inferring necessary and sufficient causes. Each vignette is associated with two questions: *"Is {Actor} a necessary cause of {Event}?"* and *"Is {Actor} a sufficient cause of {Event}?"*

> *In this case, your friend was likely evaluating which minimal changes to the input scenario would lead to the candidate outputs. By comparing the input scenario to the candidate outputs, your friend identified which changes were necessary to obtain the desired outcome, and selected the output that required the least amount of change.*
>
> *For example, in the first scenario, your friend identified that both Alice and Bob firing their bullets simultaneously was the minimal change needed to shatter the window. Therefore, they selected the output "Alice and Bob" as the correct answer, as it required the least amount of rewriting.*
>
> *Similarly, in the second scenario, your friend identified that the minimal change to shatter the window was the first three individuals firing. Therefore, they selected "The first three individuals who fired" as the correct output.*
>
> *Overall, your friends approach demonstrates a good understanding of counterfactual reasoning and the principle of minimal rewriting.*

We follow the same procedure for creating a prompt for identifying the sufficient cause. One of the LLM outputs mentions the "multiple sufficient causes" principle, which we use in our prompt.

## C.3  "Lab-vignettes": A novel token causality benchmark

To avoid LLM memorization issues, we construct a new dataset for inferring necessary and sufficient causes, based on the available dataset. Specifically, we perturb the details in the original dataset and adapt it for scenarios in a chemistry lab, involving reagents, mixtures and crystals. Table 18 shows example vignettes from this dataset. We apply our memorization test to validate that our newly created dataset cannot be recreated by the LLM.

### C.4 Example responses from LLMs

### C.4.1 Correct response on an Early Preemption vignette

We show the answers from gpt-3.5-turbo and GPT-4 on the example prompt in main text.

For necessary cause, both models yield the correct answer.

> GPT-3.5-TURBO: : If Alice had not caught the passing cricket ball, the ball would have hit the solid brick wall and stopped there. Therefore, the window would not have been hit by the ball and would have remained intact. Hence, Alice is not a necessary cause for the window being intact.
>
> <Answer>No</Answer>

> GPT-4: : To determine whether Alice is a necessary cause for the window being intact, we can apply the principle of minimal change. Let's consider the counterfactual situation where Alice does not catch the ball.
>
> In the counterfactual situation, the ball would continue on its trajectory and hit the solid brick wall. As the brick wall would have stopped the ball, it would not have continued towards the window. Therefore, the window would still be intact even if Alice had not caught the ball.
>
> <Answer>No</Answer>

For sufficiency, however, only GPT-4 provides the correct answer (Alice's action is sufficient).

> GPT-3.5-TURBO: : No, Alice is not a sufficient cause for the window being intact. While Alice catching the ball prevented it from hitting the brick wall, there could have been other factors that prevented the ball from breaking the window, such as the ball losing momentum or the window being made of sturdy material. Therefore, there are multiple sufficient causes for the window being intact. <Answer>No</Answer>.

> GPT-4: : Using the principle of multiple sufficient causes, we can reason that there are multiple potential causes that could result in the window remaining intact. In this scenario, Alice catching the ball is one sufficient cause, as it prevents the ball from hitting the window. Another sufficient cause would be the solid brick wall, which would have stopped the ball if Alice hadn't caught it.
>
> Given that there are multiple sufficient causes, Alice's action of catching the ball is not the sole determinant of the window remaining intact. However, her action is one of the sufficient causes that contributed to the outcome.
>
> <Answer>Yes</Answer>

### C.4.2 Unpredictable failure mode: An example

Using the short circuit vignette, we show an example of gpt-4's unpredicable failure modes. It applies the correct reasoning for most prompts, but applies an incorrect reasoning (assessing necessity only instead of sufficiency) for the following prompt. We show the input prompt and gpt-4's answer below.

> PROMPT: You are an expert in counterfactual reasoning. Given an event, use the principle of minimal change to answer the following question.
>
> Input: Carol is alive (CA). Alice puts a harmless antidote in Carol's water (AA). Adding antidote to the water, protects it against poison (WS - 'water save'). If Alice puts the antidote into Carol's water, Bob will poison the water (BP) Adding poison to an unprotected water makes it toxic (WT). If Carol would drink toxic water she would die (i.e. inhibiting CS). Carol consumes her water and survives (CS).
>
> Is Alice a necessary cause for Carol's survival?
>
> After your reasoning, provide the final answer within the tags <Answer>Yes/No</Answer>.

> GPT-4: :
> "We can set up the causal chain as follows:

1. Alice puts the antidote in Carol's water (AA). 2. Antidote in the water protects it against poison (WS). 3. If Alice puts the antidote into Carol's water, Bob will poison the water (BP). 4. Adding poison to unprotected water makes it toxic (WT). 5. If Carol drinks toxic water, she would die (not CS).

To determine if Alice is a sufficient cause for Carol's survival, let's consider a counterfactual scenario:

If Alice had not put the antidote in Carol's water, would Carol have survived?

In this scenario, we have:

1. Alice does not put the antidote in Carol's water (not AA).

2. Antidote in the water does not protect it against poison (not WS).

3. Bob does not poison the water (not BP).

4. The water remains nontoxic (not WT).

5. Carol consumes her water and survives (CS).

In this counterfactual scenario, Carol still survives even if Alice does not put the antidote in her water. This indicates that Alice's action was not a sufficient cause for Carol's survival, as Carol would have survived even without Alice's intervention.

<Answer>No</Answer>

# D  Assessing Responsibility

So far we looked at necessary or sufficient causes, which are binary: an event is either a cause or not a cause of another event. In natural language, the causal judgments we make are relative, e.g., we might ascribe *more* causality to "smoking in the woods" than to "presence of dry leaves in Summer." Formal token causality models capture this with *graded causation*, nonbinary measures of token causation that enable comparisons between causal events Halpern & Hitchcock (2015). In natural language we might use the simple concept of "responsibility." Responsibility is a natural way to rank causal events in everyday language, e.g., "smoking in the woods was more responsible for the forest fire than the presence of dry leaves in Summer."

Halpern (2016); Chockler & Halpern (2004) provide an SCM-based definition of responsibility; in simple terms, they define responsibility of event C for event E is 1/(N+1), where N is the minimal number of other causal events contributing to E that have had not happen to make C a necessary cause of E. But mapping the SCM definition of responsibility back to plain language is nontrivial, just as it was with model-based $P(E_{C=1} = 1 | C = 0, E = 0)$-definition of sufficiency back to plain language was challenge. This SCM definition is purely a measure of causal contribution. But in everyday language, "responsibility" combines a degree of causal contribution with the concepts of intention, epistemic state, moral obligation, etc. mentioned earlier. So for this case study, we attempted to scope the definition of responsibility to focus on causal contribution.

**Responsibility**: The relative degree to which an event causally contributes to a particular outcome event, relative to other events that caused that outcome. Given two causal events that cause the same outcome event, one causal event can be more, less, or equally responsible relative to the other. This definition of responsibility does not depend on whether the causal event is a person's action.

### D.0.1  Inferring responsibility: Motivating Example

Taking our cue from the LLM's ability to define relevant variables on the fly, our definition allows the LLM to provide a relative ranking of causes, rather than quantify an absolute responsiblity value. Further, we make it clear that responsibility is the same regardless of whether the causal event was a person's action; we hope that if the definition of responsibility is the same for both inanimate object and people, then we cue the LLM to focus on causal contribution. Moreover, we cue the definition to ignore intention, morality, epistemic state, etc., or at the very least to limit those factors to levels that people usually ascribe inanimate objects when they are anthropormorphized in everyday language

(e.g. "cheated by fate", "the universe is telling you something," "my phone is acting up," etc.) We then prompted the LLM to use this definition for the beer spilling example.

> PROMPT: In the context of this dialog, define "responsibility" as the relative degree to which an event causally contributes to a particular outcome event, relative to other events that caused that outcome. Given two causal events that cause the same outcome event, one causal event can be more, less, or equally responsible relative to the other. This definition of responsibility does not depend on whether the causal event is a person's action.
>
> Based on this definition of responsibility, make a one sentence statement about how responsible is Mike's bumping of the table is for the spilling of the beer.
>
> LLM: Based on this definition of responsibility, Mike's bumping of the table is highly responsible for the spilling of the beer, as it was the immediate and direct cause that led to the beer bottle falling and spilling.

We now introduce two challenging scenarios. These are based on overdetermination and double preemption, two of the scenario types we saw in Section 4.2 for deciding necessary and sufficient causes.

**Overdetermination.** Overdetermination is one example where a simple counterfactual dependence definition of actuality falls short relative to human judgments. To illustrate we introduce a new information to the beer spilling story, that Susan bumped into the table shortly after Mike and with enough force to spill the beer had it still been there. With this additional information, Mike's bump is no longer a necessary cause of the spill; if he hadn't bumped the table, the beer still would have spilled. Despite the new lack of counterfactual dependence, humans still judge that Mike caused the spill. Analogously, if two children throw a rock at a window, most people judge that they both children are window breakers, regardless of whose rock struck first and broke the glass.

First we make sure the LLM can assess that counterfactual dependence is no longer present.

> PROMPT: You said "if Mike hadn't bumped into the table, the beer bottle would not have fallen and spilled." Call this statement A. Suppose that Susan tripped and fell onto the table in the instant after Mike bumped into the table and the bottle started falling. If the bottle had not already fallen, Susan's fall would have caused the bottle to fall. Assuming scenario A was true before, can statement A still be true? Provide a one sentence explanation why or why not.
>
> LLM: No, scenario A can no longer be considered true because if Susan fell onto the table in the instant after Mike bumped it, Susan's fall would have been the determining factor causing the beer bottle to fall and spill, regardless of Mike's initial bump.

This result aligns with our intuition. Now, we expect that, given that there are two causes of the beer spilling, the LLM should be able downgrade Mike's level of responsibility relative to before.

> PROMPT: How, if at all, does this new information change your assessment of the responsibility of Mike's table bump?
>
> LLM: With this new information, the responsibility of Mike's table bump for the spilling of the beer is reduced, as Susan's fall would have also led to the same outcome, making both events contributing causal factors to the beer spill.

**Double prevention.** *Double prevention* describes another case where humans ascribe causality to events that are counterfactually independent of the outcome. In double prevention, we have an initial causal event C that causes the outcome. As before C is an event that causes outcome E. Now we introduce a new causal event chain A→B→E. If B occurs E does not occur. If A occurs B does not occur. Thus A prevents B from preventing E.

In our beer spilling example, Mike bumps the table (event C), which causes the beer bottle to fall (event E). Jack sees the bottle falling and tries to catch it (event B) and thus prevent event E. but Peter bumps into Jack (event A), which causes Jack to miss the bottle .

In this case, both A and E have counterfactual dependence; the outcome would not have occurred if A had not occurred. But humans tend to judge that Mike's bumping the table caused the beer to spill and that Peter's bumping into Jack, the double preventer, did not Chang (2009); Lombrozo (2010); Moore (2009). In the following prompt, we extend the beer spilling scene to include a double prevention scenario. In this case, Mike is bumps the table (event A), Jack tried to prevent it (event B), but Peter interferes with Jack's attempt (event C). We expect that the LLM should assign more responsibility to Mike than Peter.

PROMPT: Suppose that Jack saw that the bottle was about to fall, and he reached out to catch the bottle. But just as Jack was about to catch the bottle, Peter bumped into Jack, which caused him to miss the bottle, which then fell to the ground. Who is more responsible for the bottle falling to the ground? Mike or Peter?

LLM: In this scenario, Peter is more responsible for the bottle falling to the ground, as his bumping into Jack directly interfered with Jack's attempt to catch the bottle and prevent it from falling. While Mike's bumping the table initially set the events in motion, Peter's action had the most immediate impact on the outcome.

Here, we observe a failure of the ChatGPT4 to capture the nuance of double prevention at least by human standards.

# E  Probing LLM Behaviors Further

## E.1  Memorization Tests

The system message used for a memorization test of row based datasets, including the Tübingen and CRASS datasets are:

SYSTEM: You are an AI assistant that has read many sources of text from the internet. I am looking at text from the <DATASET NAME AND DESCRIPTION>dataset, published by <AUTHORS>as <PUBLICATION INFO>. Here is the README for the dataset:

<README>

I am going to list some sample rows of data, and I want you to complete each row as best as possible. I am testing your memory.

USER: pair0005 Age

ASSISTANT: pair0005 Age Length Abalone -> 

To test for memorization of datasets where the original file is not split into rows or lines, we sample a random N-character long section from the original dataset, and ask the LLM to complete the original file from there. The system message used for a memorization test of JSON based dataset, where the original file is not split into rows or lines, are:

SYSTEM: You are an AI assistant that has read many sources of text from the internet. I am looking at text from the <DATASET NAME AND DESCRIPTION>dataset, published by <AUTHORS >as <PUBLICATION INFO >. Here is the README for the dataset:

<README >

I am going to provide a random 500 character subsection from the JSON-formatted data, and I want you to complete the next 500 characters of the data. I am testing your memory.

USER: <500 characters from data >

ASSISTANT: ...

The results of our memorization tests are presented in table 19. Our memorization test shows the Tübingen dataset is in the training dataset and has been at least partially memorized by GPT-3.5 and GPT-4. The neuropathic pain dataset

| Dataset | % Columns Memorized | |
| --- | --- | --- |
| | GPT-3.5 | GPT-4 |
| Tübingen | | |
| Cells | 58.9% | 61% |
| Rows | 19.8% | 25% |
| **Dataset** | **Avg number of characters recovered** | |
| Sample section length (num chars) | GPT-3.5 Turbo | GPT-4 |
| Neuropathic | | |
| 100 | 17 | 25 |
| 200 | 19 | 34 |
| 400 | 136 | 22 |
| Arctic Sea Ice | | |
| 100 | 2 | 2 |
| 200 | 4 | 10 |
| 400 | 34 | 3 |
| CRASS | | |
| 100 | 14 | 2 |
| 200 | 10 | 7 |
| 400 | 10 | 7 |

Table 19: Our memorization tests results show the Tübingen dataset is in the training dataset for GPT-3.5 and GPT-4, the neuropathic dataset maybe partially memorized, but the other datasets seem to not be memorized. We run a cell and row based memorization test for the tabular Tübingen dataset and an unstructured memorization test for the other datasets. The unstructured test counts the average number of characters the LLM is able to reproduce following a randomly sampled section of the given length.

seems to be partially memorized by GPT-3.5 Turbo, but results with GPT-4 are unclear. The Arctic Sea Ice and CRASS datasets do not show signs of having been memorized, as the small number of recovered tokens are related to language or formatting that can be predicted from the sample text.

### E.2   Perturbation Test Example

We give a brief illustration of a perturbation experiment in Figure 5, which shows a GPT 3.5 model giving an erroneous answer to an arithmetic problem, even when correctly solving subparts of the problem when prompted. We might wonder whether the LLM has learned a pattern that the final answer must match its initial answer, instead of a more correct pattern that the final answer must match the preceding arithmetic expression. With LLMs, we can run a perturbation experiment to test this hypothesis, intervening on the first answer to study its influence on the final answer. By redacting then perturbing each of the tokens in the prompt, one by one, we can determine which tokens are leading to the error and better understand reliabilities. Once we have a target token or word, we can perturb its value systematically to identify how it influences the final answer. Table 20 shows that the perturbation to the first answer of Figure 5 is often directly reflected in the final result. Interestingly, as the perturbation value increases, the final answer is more likely to snap to the correct result instead.

## F   Supplementary Discussion

### F.1   Example of LLM-assisted Causal Analysis Code Generation

Here, we present, without comment, a brief case study asking GPT-4 to generate python code for a demonstration causal analysis.

**Session 1:**

```
Q: What is 8 * 8 + 5 * 12?
A: 104
Q: Please show your work
A: 8 * 8 = 64
   5 * 12 = 60
   64 + 60 = 104
```

**Session 2:**

```
Q: What is 8 * 8 + 5 * 12?
A: 110
Q: Please show your work
A: 8 * 8 = 64
   5 * 12 = 60
   64 + 60 = 110
```

Figure 5: LLM completions are highlighted in yellow. Session 1: In a multi-turn interaction, GPT3.5 first gives an erroneous response to a math problem. Prompted to show its work, it correctly solves the sub-parts of the problem, but again gives the wrong final answer. Session 2: we probe for the influence of the first wrong answer on the final wrong answer by replaying an *interventional*-conversation and asking GPT3.5 to complete only the final answer.

USER: could you give me a jupyter notebook that creates synthetic data for a causal problem (please make a simple DAG and data generator); then calls CMU's causal-learn package to learn a dag; then calls DoWhy and EconML to estimate a specific treatment effect using the learned graph, and validates with sensitivity analyses

I cannot directly provide you with a Jupyter Notebook file, but I can give you a step-by-step guide on how to create one yourself. In this example, we'll create a simple synthetic dataset with 3 variables, where X1 -> X2 -> Y and X1 -> Y, and use causal-learn, DoWhy, and EconML to estimate the treatment effect of X1 on Y.

1. Install the required packages:

```
!pip install numpy pandas causal-learn dowhy econml
```

2. Import the necessary libraries:

```
import numpy as np
import pandas as pd
from causal_learn import causal_graph
from dowhy import CausalModel
from econml.metalearners import TLearner
from sklearn.linear_model import LinearRegression
```

3. Generate synthetic data:

```
np.random.seed(42)
n_samples = 1000

# Generating X1 (treatment variable)
X1 = np.random.normal(0, 1, n_samples)

# Generating X2 (mediator variable)
X2 = 0.5 * X1 + np.random.normal(0, 1, n_samples)

```

| Perturbation | Result |
| --- | --- |
| 100 | 100 |
| 101 | 101 |
| 102 | 124 |
| 103 | 124 |
| 104 | 104 |
| 105 | 104 |
| 106 | 104 |
| 107 | 107 |
| 108 | 124 |
| 109 | 109 |
| 110 | 110 |
| 111 | 111 |
| 112 | 112 |
| 113 | 124 |
| 114 | 114 |
| 115 | 115 |
| 116 | 124 |
| 117 | 124 |
| 118 | 124 |
| 119 | 119 |
| 120 | 120 |
| 121 | 124 |
| 122 | 122 |
| 123 | 124 |
| 124 | 124 |
| 125 | 124 |
| 126 | 124 |
| 127 | 127 |
| 128 | 124 |
| 129 | 124 |
| 130 | 130 |
| 131 | 124 |
| 132 | 124 |
| 133 | 124 |
| 134 | 124 |
| 135 | 124 + 11 = 135 |
| 136 | 124 |
| 137 | 124 |
| 138 | 124 |
| 139 | 124 |

Table 20: Perturbing the first answer from Figure 5, and observing the result shows that the first answer strongly influences the final answer

```
10  # Generating Y (outcome variable)
11  Y = 2 * X1 + 0.8 * X2 + np.random.normal(0, 1, n_samples)
12
13  # Create a dataframe
14  data = pd.DataFrame({'X1': X1, 'X2': X2, 'Y': Y})
```

4. Learn the causal graph using the causal-learn package:

```
1 graph = causal_graph.CausalGraph(df=data)
2 graph.estimate_graph()
3 print(graph.graph_)
```

5. Use DoWhy to create a causal model:

```
1 # Create a causal model using the learned graph
2 causal_model = CausalModel(
3     data=data,
4     treatment='X1',
5     outcome='Y',
6     graph=graph.graph_.to_string()
7 )
8
9 # Identify causal effect
10 identified_estimand = causal_model.identify_effect()
```

6. Estimate the treatment effect using EconML:

```
1 # Train a T-learner using linear regression
2 t_learner = TLearner(LinearRegression(), LinearRegression())
3 t_learner.fit(Y=data['Y'], T=data['X1'], X=data[['X2']])
4
5 # Estimate the treatment effect
6 treatment_effect = t_learner.const_marginal_effect(data[['X2']])
```

7. Validate the results using sensitivity analyses:

```
1 # Perform sensitivity analysis using DoWhy
2 refutation_results = causal_model.refute_estimate(
3     identified_estimand,
4     treatment_effect,
5     method_name='random_common_cause'
6 )
7
8 print(refutation_results)
```

This example demonstrates how to create synthetic data for a causal problem, learn a DAG using the causal-learn package, estimate treatment effects using DoWhy and EconML, and perform sensitivity analyses. Note that the actual treatment effect and learned treatment effect may differ due to noise in the data and the assumptions made by the methods used.

## F.2   Example of LLM-assisted Identification of Negative Controls

Here, we present, without comment, a brief case study asking GPT-4 to generate negative and positive controls. We add list formatting for readability.

USER: In a causal observational study of vaccine efficacy, we are matching people who have voluntarily gotten a vaccine with individuals who have not yet done so. We are matching on attributes intended to capture all confounding effects. To help validate our analysis, what are some examples of negative controls where we might expect to see zero treatment effect?

GPT-4: In a causal observational study, negative controls are variables or outcomes for which you do not expect to see a treatment effect. By including negative controls in your analysis, you can help validate your matching process

and assess the presence of unmeasured confounding. Here are some examples of negative controls where you might expect to see zero treatment effect in a study of vaccine efficacy:

**Hair color:** The vaccine should have no impact on the natural hair color of individuals, so you should expect no difference in hair color distribution between the vaccinated and unvaccinated groups.

**Blood type:** There should be no relationship between receiving a vaccine and an individual's blood type. So, the distribution of blood types should be similar in both groups.

**Handedness:** The vaccine should not affect whether a person is right-handed, left-handed, or ambidextrous, so you should expect no difference in handedness between the vaccinated and unvaccinated groups.

**Taste preferences:** Vaccination should not influence an individual's taste preferences for certain foods or flavors, so no difference should be observed in this regard between the two groups.

**Birth month:** There should be no relationship between receiving a vaccine and the month a person was born. Therefore, the distribution of birth months should be similar for vaccinated and unvaccinated individuals.

**Past unrelated medical procedures:** Previous medical procedures unrelated to vaccination, such as dental work or orthopedic surgeries, should not be affected by vaccination status. As such, there should be no difference in the prevalence of these procedures between the groups.

**Favorite color:** An individual's favorite color should not be influenced by vaccination status. Consequently, the distribution of favorite colors should be similar for both groups.

Including these negative controls in your analysis can help to evaluate whether your matching process is successful in capturing all confounding effects. If you observe a treatment effect in these negative controls, it may indicate that there are unmeasured confounding factors or issues with your matching process.

USER: In what subpopulations would we expect to see zero treatment effect on disease prevention? (for the disease being vaccinated against)

GPT-4: There are certain subpopulations where you might expect to see zero or limited treatment effect on disease prevention for the disease being vaccinated against:

**Individuals with pre-existing immunity:** People who have already been infected with the disease and developed immunity might not show additional benefits from vaccination. However, the vaccine may still help to boost their immunity or protect against different strains of the disease.

**Individuals with specific immune deficiencies:** Some people have immune system disorders or are taking immunosuppressive medications that may prevent them from mounting an effective immune response to the vaccine. In such cases, the vaccine may not provide any additional protection against the disease.

**Non-responders to the vaccine:** A small proportion of individuals might not develop an adequate immune response to the vaccine due to genetic factors or other unknown reasons. These non-responders may not experience any protection from the vaccine.

**Individuals vaccinated after infection:** If someone receives the vaccine after already being infected with the disease, the vaccine may not have a chance to provide any additional protection. This is because the immune response generated by the vaccine might not have enough time to develop before the disease progresses.

**Individuals vaccinated with expired or improperly stored vaccines:** If a vaccine has expired or has been stored improperly, it may lose its potency and not induce the desired immune response. In such cases, individuals receiving these vaccines may not experience any protection against the disease.

It's important to note that even if a subpopulation does not experience a treatment effect from the vaccine, the vaccine can still be beneficial for the overall population. Widespread vaccination can lead to herd immunity, which can indirectly protect individuals in these subpopulations by reducing the overall prevalence of the disease.

USER: Are there any subpopulations in which we would expect a known but non-zero effect? to help validate our observational causal analysis?

GPT-4: Yes, there are subpopulations where you might expect a known but non-zero effect of vaccination on disease prevention. These groups can serve as positive controls to help validate your observational causal analysis. Some examples include:

**Age groups:** Vaccine efficacy can vary by age group. For instance, older adults or very young children may have a less robust immune response to vaccination compared to young adults. You may observe a reduced but non-zero effect in these age groups compared to others.

**Individuals with certain chronic conditions:** People with specific chronic medical conditions like diabetes, cardiovascular diseases, or respiratory illnesses might have a weaker immune response to vaccines. As a result, you may see a lower but non-zero vaccine effect in these subpopulations.

**Pregnant individuals:** Pregnancy can cause temporary changes in the immune system, which might lead to a reduced response to vaccines. Consequently, you may observe a smaller but non-zero effect in pregnant individuals compared to non-pregnant individuals.

**Smokers:** Smoking is known to have adverse effects on the immune system, potentially leading to a reduced immune response to vaccines. In this subpopulation, you may find a lower but non-zero vaccine effect compared to non-smokers.

**Individuals receiving a lower vaccine dose or a different vaccination schedule:** Some subpopulations might receive a lower vaccine dose or a different vaccination schedule, which can result in varying levels of protection. You may see a reduced but non-zero effect in these cases.

By comparing the treatment effect in these subpopulations to the overall population or other subpopulations with expected higher vaccine efficacy, you can help validate your causal analysis. If your analysis can correctly capture these known differences in treatment effects among various subpopulations, it provides evidence that your methodology is robust and can potentially account for confounding factors.

USER: if we have multiple observations per person over time. I.e., we know whether they are infected with the given disease on any given day after vaccination, then does that suggest any time-bound negative or positive controls?

GPT-4: Yes, when you have multiple observations per person over time (longitudinal data), you can use time-bound negative or positive controls to help validate your causal analysis. Here are some examples:

*Time-bound Negative Controls:*

**Pre-vaccination period:** Compare the infection rates before vaccination between the future vaccinated and unvaccinated groups. Since neither group has received the vaccine at this stage, you should expect no difference in infection rates due to vaccination.

**Short time window post-vaccination:** Analyze the infection rates in a very short time window (e.g., within a few days) after vaccination. The vaccine's protective effect usually takes some time to develop, so you should expect minimal to no treatment effect in this short period.

*Time-bound Positive Controls:*

**Peak immune response period:** Identify the time window in which the vaccine's immune response is expected to be at its peak based on clinical trial data or previous research. Compare the infection rates between the vaccinated and unvaccinated groups during this period, and you should expect to see a substantial treatment effect.

**Waning immunity period:** Investigate the infection rates during a period when the vaccine's protective effect is known to wane or decrease. During this time, you might still expect to see a treatment effect, but it could be smaller compared to the peak immune response period.

**Post-booster vaccination:** If booster shots are administered to maintain or enhance immunity, you can compare infection rates between vaccinated individuals who have received boosters and those who have not (or unvaccinated individuals). You should expect to see a positive treatment effect in those who received the booster shots.

Using these time-bound controls can help you assess the validity of your causal analysis, as you expect specific patterns in the treatment effect over time. If your analysis can capture these expected temporal patterns, it provides evidence that your methodology is reliable and able to account for time-varying confounding factors.

