# OpenReview forum: "Causal Reasoning and Large Language Models: Opening a New Frontier for Causality"
_TMLR — Accepted by TMLR_

### Review · Reviewer_i2Bg · 2024-03-29

**Summary Of Contributions:**

A lengthy and extensive study of LLMs for various causal inference tasks.

**Audience:**

Yes

**Claims And Evidence:**

Yes

**Requested Changes:**

- A memorization test for every dataset in the paper: Neuropathic pain, NHD, CRASS, Kueffner 2021, BIG-Bench Hard and any other I may have missed - critical
- Discussion of overfitting, prompt selection on the test set, etc - critical
- Update discussion in 5.1 pending response to this review - critical

**Strengths And Weaknesses:**

Thanks to the authors for the obvious very hard work on this paper. I think it's a great contribution to the field, but I have some substantial concerns that I outline below. But make no mistake - I believe this paper is very worthwhile!

Strengths
------------
- A ton of empirical work and careful thinking
- Lots of experiments
- Clear writing

Weaknesses
----------------
- The biggest concern is that LLMs have seen the various datasets in this paper. If they have, the results are far from conclusive. As you have made your own "lab-vignettes" dataset, I strongly recommend (a) testing whether multiple LLMs have memorized each dataset you use (not just Tubingen) and (b) making a new dataset for each of those that have been memorized.

- 3.1.1 - There are only 108 questions so overfitting is a serious concern. Did you choose your prompts once without any tuning? Please detail how the process of prompt tuning was structured, and discuss overfiting risk. Similarly for for Table 4: there are multiple GPT-4 entries in the table (minor: one is incorrectly spelled), with widely different scores. If you (hypothetically) kept modifying the prompts/approaches while looking at the test set accuracy, then leakage is likely. Please clarify how prompt tuning was done.

- 3.2.1 - The same questions apply. How many prompts did you try before you found the one where you add "C: No causal relationship exists”?

- 3.3 - These results are both *very* concerning and confusingly presented. Since the LLM has *seen* the labels (as per your test), then I don't see how the basic claim of your paper (that LLMs are good at causality) is being truly tested. This is just plain overfitting to the test set, and the comparisons to other non-memorizing algorithms are unfair. Please repeat the memorization test for all the datasets tested (Neuropathic pain, NHD, CRASS, Kueffner 2021, BIG-Bench Hard causal judgment), and report the results.

- 4.1 - Related to the previous point, there are a number of claims about LLM capability in this paper that resemble "The results indicate that large language models of GPT-3.5 and 4 series represent a substantial jump in LLM ability to generate counterfactual arguments". Before making these claims, you should clearly establish that LLMs have not memorized the dataset.

- 4.2.1 - "Since the story was published in 2022, it cannot have been in the training set of either GPT 3 or GPT 4 models." Which exact LLM version is being used here?

- Page 23 - "This is remarkable because the LLM was not explicitly provided definitions of necessary and sufficient causes" - why not provide these as well?

- 5.1 - Generally speaking, benchmarks that have been created for purely numerical methods for causal discovery are "valid" (colloquial meaning) because the numerical methods don't have any prior information about the meaning of the variables. So they have to "figure out" causality from scratch (with some model structure priors built in by expert humans). This is important because we generally care about inferring causality in novel situations. From the perspective of the numerical causal discovery method, then, each benchmark dataset *is* a novel situation.  I don't fully believe this because of a field's tendency to overfit to its benchmarks, but that is the ideal. The fact that LLMs learned causal structure from their extensive exposure to terabytes of human text is a good finding, but that does *not* suggest they are useful for realistic causal discovery (which is what you propose in 5.1). This is because the causal questions of practical interest are new ones. I.e., not ones that appear somewhere in the pre-training corpus of a modern LLM. You have shown (subject to the concerns above) that LLMs can perform "pairwise causal discovery", "counterfactual reasoning task", and "event causality". If solving these tasks requires extant human knowledge, then it makes sense that powerful LLMs can perform these tasks to some degree. However, can LLMs perform these tasks in *novel* situations? Numerical algorithms can to some degree - we have evidence of this because they are able to perform well without strong knowledge-based priors in existing benchmarks. But what can we say about LLM causal inference for extra-training corpus scenarios?

---

### Review · Reviewer_ePfk · 2024-04-02

**Summary Of Contributions:**

This work presents an investigation of causal reasoning in large language models. The results suggest that the best performing models (GPT-3.5 and GPT-4) are able to answer general causal knowledge questions (i.e., stating which of two real-world variables is likely to have a causal effect on the other), demonstrate some capacity for counterfactual reasoning, and are able to assess causal necessity and sufficiency. The implications of these results for causal modeling approaches is discussed.

**Audience:**

Yes

**Broader Impact Concerns:**

Broader impacts and ethical implications are adequately discussed.

**Claims And Evidence:**

No

**Requested Changes:**

- Some of the experiments (e.g., the cause-effect task) need to performed with datasets that are either novel, or that we can be reasonably confident aren't in the training data.
- The potential causal link between the presence of the datasets in the training data and the ability of the models to solve the tasks presented in this paper should be discussed.
- Chain-of-thought should be performed before a final answer is provided, rather than as a justification of an answer already provided.
- The relationship of the present experiments to previous work should be more clearly explained.

**Strengths And Weaknesses:**

## Strengths
- Background literature and concepts in causal reasoning are clearly explained
- Experiments are performed across a range of LLMs
- Quantitative results are accompanied with qualitative analyses providing a richer picture of the way in which these models are solving causal reasoning tasks
- There is a thoughtful discussion of the ways in which LLMs might be integrated with more traditional causal reasoning techniques

## Weaknesses
- The primary weakness is that many of the datasets are likely present in the training data, as demonstrated by the memorization analyses. It is good to include these analyses, but the results do raise the possibility that the LLMs are perhaps not able to perform causal reasoning in a generalizable way. It would be ideal to combine these results with results from a newly developed dataset, or one that we can be reasonably confident isn't in the training data. This was done for one of the experiments, but the possibility still remains that the performance observed in the other experiments is dependent on similarity to the training data. It would also be good to discuss the ways in which the presence of these datasets in the training data may or may not explain the performance of the models on these tasks (i.e., the mere presence of this dataset in the training data doesn't automatically explain the ability of the LLM to perform the task, especially if the specific task and formatting differs from the standard format of the dataset, but it is important to discuss these issues).
- The paper presents a thorough background on the relevant literature, but it is sometimes less clear how the present work is related to previous work. It would be helpful to explicitly identify the novel contributions of this work relative to previous work.
- I don't know if the tasks in the first part of the paper would really be considered causal *reasoning*. They seem more like tests of general causal knowledge. It would be good to have some discussion of how this relates to causal reasoning.
- In some of the tests on necessity and sufficiency, the model is prompted to produce a chain-of-thought explanation *after* first providing an answer (e.g., as seen in the example in section 4.2.1). This likely prevents the model from using this chain-of-thought output to reason about the task (and potentially arrive at a better answer). It would be informative to prompt the model to reverse the order of these.

## Minor notes
- The ability of models beyond davinci-002 to solve the cause-effect task is attributed to model size, but there are other factors that differ between these models, including the training data (e.g., some are trained on code) and training objective (e.g., RLHF).
- For the cause-effect dataset, weighted accuracy is reported. How are the weights determined?
- There is a typo at the top of page 16 ('geneation').

---

### Review · Reviewer_6N5S · 2024-05-19

**Summary Of Contributions:**

This paper performs robustness checks across tasks and shows that large language models might be used by human domain experts to save effort in setting up a causal analysis.

**Audience:**

Yes

**Claims And Evidence:**

No

**Requested Changes:**

The authors need to add more theory and methodological work to the paper.

**Strengths And Weaknesses:**

Strengths: The topic of causal reasoning and large language models is interesting.
Weaknesses: The work lacks theory and methodological work.

---

### Comment · Editors_In_Chief · 2025-12-02

Congratulations to the authors on this paper being named a 2025 Outstanding Certification Finalist!

For more information, see https://medium.com/@TmlrOrg/announcing-the-2025-tmlr-outstanding-certification-e26d548ff011.

---

### Decision · Action_Editor_tNxW · 2024-07-17

**Recommendation:** Accept as is

**Comment:**

The decision is mainly based on the consensus that the paper provides a thorough empirical evaluation of the causal reasoning capabilities of LLMs, which is quite novel and interesting. The reviewers appreciated the authors' thorough response to concerns about overfitting and dataset memorization.

The paper is recommended for featured certification due to its significant empirical contributions and the novelty of integrating LLMs with causal reasoning tasks. The work stands out for its potential impact on both the academic community and practical applications in various domains where causal analysis is critical.

In summary, the decision to accept the paper is supported by the reviewers' positive feedback and the authors' effective responses to concerns, demonstrating the robustness and relevance of their findings in advancing the understanding of LLMs in causal reasoning.

**Audience:**

General interest.

**Claims And Evidence:**

This paper investigates the causal reasoning abilities of large language models (LLMs). The study demonstrates that these models can generate accurate causal arguments, outperforming existing methods on tasks such as causal discovery, counterfactual reasoning, and event causality. Despite their high performance, LLMs exhibit unpredictable errors. Robustness checks suggest that these capabilities aren't solely due to memorization. Concerns about overfitting and dataset memorization remain. Reviewers generally recommend acceptance, praising the paper's extensive empirical work and clear writing, but suggesting additional theoretical and methodological enhancements. The paper highlights LLMs' potential to assist human experts in causal analysis, marking a significant step forward in integrating LLMs with traditional causal methods.